# The GF Convection Parameterization: recent developments, extensions, and applications

Saulo R. Freitas[1,2], Georg A. Grell[3], and Haiqin Li[3,4]

[1]Goddard Earth Sciences Technology and Research, Universities Space Research Association, Columbia, MD, USA
[2]Global Modeling and Assimilation Office, NASA Goddard Space Flight Center, Greenbelt, MD, USA
[3]Earth Systems Research Laboratory of the National Oceanic and Atmospheric Administration, Boulder, CO, USA
[4]Cooperative Institute for Research in Environmental Sciences, University of Colorado Boulder, Boulder, CO, USA

*Correspondence to*: Saulo R. Freitas (saulo.r.freitas@nasa.gov)

**Abstract**. We detail recent developments in the GF (Grell and Freitas, 2014, Freitas et al., 2018) convection parameterization and applications. The parameterization has been expanded to a trimodal spectral size to simulate three convection modes: shallow, congestus and deep. In contrast to usual entrainment/detrainment assumptions, we assume that Probability Density Functions (PDF's) can be used to characterize the vertical mass flux profiles for the three modes, and use the PDF's to derive entrainment and detrainment rates. We also added a new closure for non-equilibrium convection that improved the simulation of the diurnal cycle of convection, with better representation of the transition from shallow to deep convection regimes over land. The transport of chemical constituents (including wet deposition) can be treated inside the GF scheme. Transport is handled in flux form and is mass conserving. Finally, the cloud microphysics has been extended to include the ice phase to simulate the conversion from liquid water to ice in updrafts with resulting additional heating release, and the melting from snow to rain.

## 1 Introduction

Convection Parameterizations (CPs) are sub-model components of atmospheric models that aim to represent the statistical effects of a sub-grid scale ensemble of convective clouds. It is necessary in models in which the spatial resolution is not sufficient to resolve the convective circulations. These parameterizations often differ fundamentally in closure assumptions and parameters used to solve the interaction problem, leading to a large spread and uncertainty in possible solutions. For some interesting review articles on convective parameterizations the reader is referred to Frank (1984), Grell (1991), Emanuel and Raymond (1992), Emanuel (1994), and Arakawa (2004). A seminal work on CPs was done by Arakawa and Schubert (1974). Following

this, new ideas were implemented, such as including stochasticism (Grell and Devenyi, 2002, Lin and Neelin, 2003), and the super parameterization approach (Grabowski and Smolarkiewicz, 1999, Randall et al., 2003), to name a few.

An additional complication is the use of convective parameterizations on so-called "gray scales," which is gaining attention rapidly (Kuell et al., 2007, Mironov 2009, Gerard et al., 2009, Yano et al., 2010, Arakawa et al., 2011, Grell and Freitas, 2014, Kwon and Hong, 2017). The original Grell and Freitas (2014, hereafter GF2014) scheme was built based on a convective parameterization developed by Grell (1993) and expanded by Grell and Devenyi (2002, hereafter GD2002) to include stochasticism by expanding the original scheme to allow for a series of different assumptions that are commonly used in convective parameterizations and that have proven to lead to large sensitivity in model simulations. In GF, scale awareness (following Arakawa et al. 2011) was added for application on "gray scales", aerosol awareness was implemented by including a Cloud Condensation Nuclei (CCN) dependence of conversion from cloud water to rainwater in addition to using an empirical approach that relates precipitation efficiency to CCN.

This version of GF is used operational in the Rapid Refresh (RAP, Benjamin et al. 2016) at the Environmental Modeling Center (EMC) at the National Center for Environmental Prediction (NCEP) of the National Weather Service (NWS) in the US, at the Global Modeling and Assimilation Office of NASA Goddard Space Flight Center, and in the Brazilian Center for Weather Forecast and Climate Studies (CPTEC/INPE). Scale awareness was further tested successfully in GF, in a nonhydrostatic global model with smoothly varying grid spacing from 50 to 3km (Fowler et al. 2016), and also in a cascade of global-scale simulations with uniform grid size spanning from 100 km to few kms using the NASA GEOS GCM (Freitas et al., 2018, 2020). The use of GF in other modeling systems and for other applications required further modifications to represent physical processes such as momentum transport, cumulus congestus clouds, modifications of cloud water detrainment, and better representation of the diurnal cycle of convection. In this paper we describe some of the new features introduced recently.

In Section 2, we will describe the new implementations, Section 3 will show some results from single column models to full 3D simulations, and Section 4 will conclude and summarize results.

## 2 New developments and extensions

### 2.1 The trimodal formulation

The original unimodal steady-state updraft deep plume has been replaced by a trimodal formulation, which allows up to three characteristic convective modes (Johnson et al., 1999): shallow, congestus, and deep. Our new approach lies in between the two extremes: just one bulk cloud, as described in Tiedtke (1989), Grell (1993), and many others, and a full, spectral cloud size approach (e.g., Arakawa and Schubert, 1974, Grell et al. (1991), Grell (1993), Moorthi and Suarez, 1992, Baba 2019). In our approach we are not claiming to represent three plumes, but PDF's characterizing plumes. For example, the PDF for deep convection is a statistical average of deep plumes in the grid box, and may include impacts from several plumes.

Each mode of our trimodal formulation is characterized by a PDF (see Section 2.2 for details) that determines average lateral mixing. For each mode we assume a characteristic initial gross lateral entrainment rate to represent an approximate size of one of the three modes of convection in the grid box. See the section 2.2 for more details about how the entrainment and detrainment rates (lateral mixing) are derived from the PDF's in GF. The deep and congestus modes are accompanied by convective scale saturated downdrafts sustained by rainfall evaporation. Associated with each mode, a set of closures to determine the mass flux at the cloud base were introduced to adequately account for the diverse regimes of convection in a given grid cell. The three modes transport momentum, tracers, water, and moist static energy. For mass and energy, the spatial discretization of the tendency equation is conservative on machine precision. The three modes are allowed to cohabit a given model grid column. The parameterization is performed over the entire spectrum executing first the shallow, next the congestus, and finally the deep mode. In this manner, the convective tendencies resulting from the development of each mode may be applied as a forcing for the next one. In this paper we will discuss results without applying feedbacks successively. The impacts of a successive application will be looked at in a future study.

### 2.1.1 Shallow convection

The source parcels for the shallow convecting plumes are defined by mixing the environmental moist static energy (MSE) and water vapor mixing ratio over a user specified depth layer (currently, the lowest atmospheric layer with 30 hPa depth). Then, an excess MSE and moisture perturbation associated with the surface fluxes is added when calculating the forcing and checking for trigger functions, as described in GF2014. The cloud base is defined by the first model level

where the source air parcel lifted from the surface without any lateral entrainment gets positive
buoyancy. Above the cloud base shallow convection growth and properties will depend much on
the PDF's that describe the vertical mass flux distribution and resulting entrainment and
detrainment rates. Since the PDF's are part of all three types of convection, the method will be
described in detail in section 2.2.  The shallow convection cloud tops are determined following
two criteria. One is by the vertical layer where the buoyancy becomes negative. The second is
defined by the first thermal inversion layer ($\partial \bar{T}/\partial z > 0$, where $\bar{T}$ is the grid scale air temperature)
above the planetary boundary layer (PBL). The effective cloud top is defined by the layer which
have the lower vertical height. The closures for the determination of the mass flux at cloud base,
suitable for shallow moist convection regime, are as follows:
– Raymond (1995), which establishes the equilibrium for the boundary layer budget of the

12       moist static energy. In this case, the flux out at the cloud base of shallow convection

13       counterbalances the flux in from surface process. This closure is called boundary layer

14       quasi-equilibrium (BLQE). The BLQE closure provides a reasonable diurnal cycle of

15       shallow convection over land as the resulting mass flux at cloud base is tightly connected

16       with the surface fluxes. The equation for the mass flux at cloud base ($m_b$) from this closure

17       reads

$$m_b = \frac{-\int_{p_s}^{p_{cb}} \frac{\partial \tilde{h}}{\partial t} \frac{dp}{g}}{(h_c - \tilde{h})_{cb}} \quad (1)$$

19       where $h_c$ and $\tilde{h}$ are the in-cloud and environmental moist static energy, respectively, g is

20       the gravity, p is the pressure, and the integral is determined from the surface to the cloud

21       base. $\tilde{h}$ is approximated by the grid-scale moist static energy and its tendency is given by

22       adding the tendencies from the grid-scale advection, diffusion in the planetary boundary

23       layer (PBL) and radiation.

24     – Grant (2001), which introduced a closure based on the boundary layer convective scale

25       vertical velocity ($w^*$) and the air density at the cloud base ($\rho$).  In this closure, $m_b$ is simply

26       given by:

$$m_b = 0.03 \ \rho w^* \quad (2)$$

28     – Rennó and Ingersoll (1996) and Souza (1999), which applied the concept of convection as

29       a natural heat engine to provide a closure for the updraft mass flux at cloud base:

$$m_b = \frac{\eta F_{in}}{T_{cape}} \quad (3)$$

where $\eta$ is the thermodynamic efficiency, $F_{in}$ is the buoyancy surface flux and $T_{cape}$ is the total convective available potential energy, which is approximated by the standard CAPE calculated from the vertical level of the air parcel source to the cloud top (Souza, 1999).

**2.1.2 Congestus and deep convection**

Congestus and deep convection share several properties and will be described together in this section. Both allow associated convective scale saturated downdraft (see Grell 1993 for further details). As for shallow convection, they are distinguished by characteristic different initial gross entrainment rate (see section 2.2) that are supposedly characterizing the average cloud size of deep and congestus convection. The cloud bases are found following the same procedure described for the shallow convection. For deep convection, the cloud top is defined by the vertical layer where the buoyancy becomes negative. For congestus convection, we proceed looking for the thermal inversion layer which is closest to the 500 hPa pressure level. The level of that inversion layer defines the cloud top for the congestus mode.

The closures formulations to determine the cloud base mass fluxes for deep convection are described in GD2002. For congestus, the closures BLQE (Eq. 1) and based on W* (Eq. 2) described in Section 2.1.1 are available, besides the instability (measured as the cloud work function) removal using a prescribed time scale of 1800 seconds (see Section 2.3 for further details).

**2.2 The use of PDF's to assume statistical representation of normalized vertical mass flux profiles**

The new version applies Beta-PDF functions to represent the average statistical mass flux of the plumes. We assume that the average normalized mass flux profiles for updrafts ($Z_u$) and the downdrafts ($Z_d$) in the grid box may be represented by a Beta function, which is given by:

$$\boldsymbol{Z_{u,d}(r_k) = c \, r_k{}^{\alpha-1} \, (1 - r_k)^{\beta-1}} \quad (4)$$

Where c is a normalization constant to assure that total probability is 1., $r_k$ is the location of the mass flux maximum, given by the ratio between the pressure depth from where the maximum of

the cloud is in relation to the cloud base related to the total depth of the cloud

$$r_k = \frac{p - p_{base}}{p_t - p_{base}} \quad (5)$$

$$c = \frac{\Gamma(\alpha+\beta)}{\Gamma(\alpha)+\Gamma(\beta)} \quad (6)$$

α and β determine the skewness of the function and $\Gamma$ is the Gamma function. In GF they depend
to a large part on where the maximum of the PDF is located. For shallow and congestus type
convection, the maximum is located towards the cloud base. For shallow convection it is assumed
to be at or just above the level of free convection. For congestus we assume this level to be higher,
at half of the congestus cloud depth. For deep convection this level is given by the level where the
stability changes sign, where the stability is given by the difference from in-cloud moist static
energy and environmental saturation moist static energy. This is equivalent to assuming that the
strong increase in static stability at those levels will – statistically – lead to an increase in
detrainment and a possible decrease in updraft radius (not necessarily updraft vertical velocity).
For deep convection we assume

$$\beta = 1.3 + \left(1. - \frac{p - p_{base}}{1200.}\right) \quad (7)$$

Then, $\alpha$ is imply given by

$$\alpha = \frac{r_{km}(\beta - 2.) + 1.}{1. - r_{km}} \quad (8)$$

Where $r_{km}$ is the value of $r_k$ at the level of maximum mass flux. $\alpha$ $and$ $\beta$ determine the skewness
of the PDF. For shallow convection we use $\beta = 2.2$, for congestus convection $\beta = 1.3$. The
downdrafts are assumed to reach maximum mass flux – in a statistical sense – at or below cloud
base, therefore

$$r_k = \frac{p_{base} - p(1)}{p_{base} - p(1)} \mathbf{(9)}$$

With $\beta = 4$.
Once the normalized mass flux profiles are defined, the entrainment and detrainment rates are
adjusted accordingly. First, initial entrainment rate is given that is meant to characterize the type
of convection in the grid box. This is assumed to be the initial rate at the cloud base. In the version
of the parameterization that is used in the *RAPid refresh hourly update cycle* at the National

Weather Service of the U.S. (hereafter RAP) and is available to the community using github, we use

$$\mathcal{E}(z) = 7.e-5, 3.e-4, 1.e-3 \quad (10a)$$

for deep, congestus, and shallow convection respectively, with

$$\delta(z) = 0.1\ \mathcal{E}(z), 0.5\ \mathcal{E}(z), 0.75\ \mathcal{E}(z) \quad (10b)$$

where $\mathcal{E}, \delta$ are the entrainment/detrainment rates (m$^{-1}$), With initial $\mathcal{E},\ \delta$ and the PDFs for $Zu$ defined, the effective lateral mixing (given through entrainment/detrainment rates $\mathcal{E}^*$ $and$ $\delta^*$), in a statistical averaged send, must be related to the vertical mass flux profiles. They are simply given by:

$$\mathcal{E}^*(z) = \begin{cases} \frac{1}{Z_u}\frac{dZ_u}{dz} + \delta(z), & z \leq z_{max} \\ \mathcal{E}(z), & z > z_{max} \end{cases} \quad (11a)$$

$$\delta^*(z) = \begin{cases} \delta(z), & z \leq z_{max} \\ -\frac{1}{Z_u}\frac{dZ_u}{dz} + \mathcal{E}(z), & z > z_{max} \end{cases} \quad (11b)$$

where z is the vertical height and $z_{max}$ is the vertical height where resides the maximum value of $Zu$. A comparison to observed mass flux profiles using a Single Column Model approach is given in Figure 4 and described later in this section in more detail.

The use of PDF's enables interesting options for introducing completely mass conserving Stochastic Parameter Perturbations (SPPs) with possibly significant increase in spread. It may of course also be used for training and tuning purposes. The operational version of the RAP uses the GF scheme with PDF's without tuning and so far without any stochastic applications. However, we are planning on using some of the approaches described next also for SPP stochastics in the near future. No shallow convection is used in the RAP, since the boundary layer parameterization (Olson et al. 2019) is used to treat shallow convection. In the next section we will describe possible ways to apply stochastics and/or use this approach for tuning.

## 2.3 Options for stochastic approaches

Following from Section 2.2, and Equation 4 we use the requirement:

$$\left.\frac{dZ_{u,d}}{dr_k}\right|_{r_k=r_{max}} = 0 \implies f(\alpha,\beta,r_{max}) = 0, \quad (12)$$

where $r_{max}$ relates to the vertical level where the mass flux profile reaches its maximum value. In this way, the function is univocally defined once $\beta$ and $r_{max}$ are specified. The two parameters $\beta$ and $r_{max}$ may be stochastically perturbed. The $r_{max}$ is used to move the level of maximum mass flux up or down, and the $\beta$ is used to define the shape of the profile. The allowed range of the beta parameter is [1, 5]. For example, Figure 1 introduces the universe of solutions for $Z_u$ of the deep convection updraft for a case where the heights of cloud base, of maximum mass flux and the cloud top are 1.2, 4.3 and 15.1 km, respectively. Choosing $\beta$ closer to 1 results in a very gentle shape of the mass flux in the troposphere, but with very sharp in/decrease at cloud base/cloud top with large entrainment/detrainment mass rates. Increasing $\beta$, the profile becomes curved and, above the level of maximum Zu, the detrainment rates dominate over the entrainment. An appropriate choice of the $\beta$ parameter implies, for example, in a more evenly detrainment of condensate water through the upper troposphere or a sharper, narrower detrainment at the very deep cloud top layer.

To give an example, using Equations 10 and 11, $z_{max}$ is defined as 4.3 km in Figure 1. Figure 2 introduces the difference between the effective entrainment and detrainment rates ($\varepsilon^* - \delta^*$) for the case shown in Figure 1. Assuming $\beta$ closer to 1 implies a very large effective entrainment/detrainment at cloud base/top with very small net mass exchange in between. As stated before, increasing $\beta$ makes the entrainment and detrainment layers wider and smoother.

The above described options for stochastically perturbing vertical mass flux distributions may of course also be used in fine tuning of model performance, in particular for operational forecasting applications. Those parameters allow slight changes in the vertical distribution of heating and drying and may be used to improve biases  in temperature and moisture profiles. As is the case with parameters and assumptions in convective parameterizations in general, the values proposed

in Section 2.2 may of course not be universal, and optimal values may need adjustments for each host model.

## 2.3 Diurnal cycle closure

Convection parameterizations based on the use of CAPE for closure and/or trigger function prove difficult in accurately representing the diurnal march of convection and precipitation associated with the diurnal surface heating in an environment of weak large-scale forcing. In nature, shallow and congestus convective plumes start a few hours after the sunrise, moistening and cooling the lower and mid-troposphere. This physical process prepares the environment for the deep penetrative and larger rainfall producing convection sets in, which usually occurs in the mid-afternoon to early evening. Models, in general, simulate a more abrupt transition, with the rainfall peaking in phase with the surface fluxes, earlier than observations reveal (Betts and Jakob, 2002). In addition to a more accurate timing of the precipitation forecast, a realistic representation of the diurnal cycle in a global model also should improve the forecast of the near-surface maximum temperature. Additionally, it improves the sub-grid scale convective transport of tracers, which should be especially relevant for carbon dioxide over vegetated areas. Moreover, as models configured in cloud-resolving scales can intrinsically capture the diurnal cycle of convection, global models with good skill on the diurnal cycle representation should yield a smoother transition from non-resolved to resolved scales. Lastly, it seems plausible that benefits on the data assimilation are also expected with a better diurnal cycle representation.

In the effort to improve the diurnal cycle in the GF scheme, we adopted a closure for non-equilibrium convection developed by Bechtold et al. (2014, hereafter B2014), which as we further demonstrate, notably improves the simulation of the diurnal cycle of convection and precipitation over land. B2014 proposed the following equation for the convective tendency for deep convection which represents the stabilization response in the closure equation for the mass flux at cloud base:

$$\left.\frac{\partial \Pi}{\partial t}\right|_{conv} = -\frac{\Pi}{\tau} + \frac{\tau_{BL}}{\tau} \left.\frac{\partial \Pi}{\partial t}\right|_{BL} \qquad (13)$$

where $\Pi$ is called density-weight buoyancy integral, and $\tau$ and $\tau_{BL}$ are appropriated time scales. The tendency of the second term on the right side of Eq. (13), is the total boundary layer production given by:

$$\left.\frac{\partial \Pi}{\partial t}\right|_{BL} = -\frac{1}{T^*} \int_{p_s}^{p_b} \left.\frac{\partial \overline{T_v}}{\partial t}\right|_{BL} dp \qquad (14)$$

where the virtual temperature tendency includes tendencies from grid-scale advection, diffusive
transport and radiation. $T^*$ is a scale temperature parameter, and the integral is performed from the
surface ($p_s$) to the cloud base ($p_b$). The justification for subracting a fraction of the boundary layer
production is that $\Pi$ already contains all the boundary layer heating but it is not totally available
for deep convection.
In GF, we follow B2014 to introduce an additional closure using the concept of the cloud work
function (CWF) available for the deep convection overturning. The CWF is calculated as

$$A = \int_{zb}^{zt} \frac{1}{c_p \bar{T}} \frac{Z_u}{1+\gamma} (h_u - \bar{h}^*)\, g dz \quad (15)$$

where, $A$ is the total updraft CWF, $z_b$ and $z_t$ are the height of the cloud base and cloud top,
respectively, $g$ is the gravity, $c_p$ the specific heat of dry air, $Z_u$ is the normalized mass flux, $\bar{T}$ is
the grid-scale air temperature, and $h_u$, $\bar{h}^*$ are the updraft and grid-scale saturated moist static
energy, respectively. The parameter $\gamma$ is given by Grell (1993, Eq. A15). Following B2014, the
boundary layer production is given by:

$$A_{BL} = \frac{\tau_{BL}}{T^*} \int_{z_{surf}}^{z_b} \left.\frac{\partial \bar{T_v}}{\partial t}\right|_{BL} g dz \quad (16)$$

where $\rho$ is the air density and the integral being performed from the surface ($z_{surf}$) to the cloud
base. From Equations 15 and 16, the available CWF ($A_{avail}$) is given by

$$A_{avail} = A - A_{BL} \quad (17)$$

and the rate of instability removal is given by $A_{avail}/\tau$ , where $\tau$ is a prescribed time scale,
currently 1 and 0.5 hour for deep and congestus modes, respectively.
While the impact for the GEOS modeling system was a substantial improvement, this may depend
on other physical parameterizations and how tendencies are applied in a GCM. For this reason, in
GF this closure is optional. On the other hand, it can be combined with any of the other closures
previously available in the scheme for deep convection.
**2.4 Inclusion of the ice phase process**
The thermodynamical equation employed in GF scheme uses the moist static energy ($h$) as a
conserved quantity for non-entraining air parcels with adiabatic displacements:

$$dh = 0 \quad (18)$$

where $h$ has the usual definition:

$$h = c_p T + gz + L_v q_v \quad (19)$$

and $c_p$ is the isobaric heat capacity of dry air, T is the temperature, g is the gravity, z is the height, $L_v$ the latent heat of vaporization, and $q_v$ the water vapor mixing ratio. However, $h$ is not conserved if the glaciation transformation occurs, and this process was not explicitly included in GF until now. Incorporating the transformation of liquid water to ice particles, Equation 18 now reads:

$$dh = L_f q_i \quad (20)$$

where $L_f$ is the latent heat of freezing, and $q_i$ is the ice mixing ratio. With the extended Equation 20, the general equation for the in-cloud moist static energy including the entraining process solved in this version of GF is

$$dh = L_f q_i + (dh)_{entr} \quad (21)$$

where $(dh)_{entr}$ represents the modification of the in-cloud moist static energy associated with the internal mixing with the entrained environmental air.

The partition between liquid and ice phases contents is represented by a smoothed Heaviside function which increases from 0 to 1 in the finite temperature range [235.16, 273.16] K, which is given by fract_liq = min(1, (max(0,(T-235.16))/(273.16-235.16))$^2$).

The melting of precipitation falling across the freezing level is represented by adding an extra term to the grid-scale moist static energy tendency:

$$\left(\frac{\partial \bar{h}}{\partial t}\right)_{melt} = -\frac{g\,L_f M}{\Delta p} \quad (22)$$

where $M$ is the mass mixing ratio of the frozen precipitation that will melt in a given model vertical layer of the pressure depth $\Delta p$.

## 3    Applications

In this section, applications associated with the features described in the previous section are discussed.

### 3.1 The trimodal characteristics revealed by single-column simulations

The GF convection scheme was implemented into the Global Model Test Bed (GMTB) Single Column Model (SCM, https://dtcenter.org/GMTB/gmtb_scm_ccpp_doc/), and SCM simulations were executed using data from the Tropical Warm Pool International Cloud Experiment (TWP-ICE, May et al., 2008) to demonstrate the trimodal characteristics and the value of using PDF's. TWP-ICE is a comprehensive field campaign that took place on January and February 2006 over Darwin, Australia.

The time series of GF simulated total (red solid), convective (red dash) and observed total
precipitation (black) are shown in Figure 3. Strong precipitation events are observed during the
active monsoon period with a major Mesoscale Convective System (MCS) on 23 January 2006
and followed by a suppressed monsoon with relatively weak rainfall (Fig. 3). 19 January 2006 –
25 January 2006 and 26 January 2006 – 02 February 2006 are defined as active monsoon and
suppressed monsoon periods for the subsequent quantitative analysis. GF captures all the peak
precipitation events during the active monsoon period. The heavy precipitation in the active
monsoon period appears underestimated, while the light precipitation events in the suppressed
monsoon period may seem overestimated. However, exact agreement cannot be expected.
Precipitation data for this data set were derived from radar data, derivation of large scale forcing
data is also not trivial. Some of this is also obvious in the calculation and discussion of the Q1 and
Q2 profiles (later in this section) The convective precipitation contributes about 78% of the total
precipitation during the active monsoon period and contributes as high as 94% of total precipitation
during the suppressed monsoon period.

15       To test the approximation of the normalized mass flux with our PDF approach, we compare
the simulated mass flux profiles with observations, as analyzed by Kumar et al. (2016). Of
particular importance for us is whether the PDF for deep convections is able to characterize deep
convective clouds in the area, since this will determine maximum entrainment and detrainment in
the GF parameterization. For completeness we also compare congestus and shallow clouds. The
mean mass flux during the whole TWP-ICE simulation period from all cumulus clouds (deep,
congestus, and shallow), shallow, congestus, and deep convection are shown in Figure 4B. The
congestus mass flux (green), which is weaker than the mass flux for deep convection, has its
maximum around 7 km height. The maximum mass flux from deep convection (red) and all
convective types (black) is around 6km and a bit under 6km, respectively.  Kumar et al. (2016)
estimated the convective mass flux from two wet season (October 2005 – April 2006 and October
2006 – April 2007) from radar observations over Darwin, Australia. Although the TWP-ICE
simulation period (19 January 2006 – 02 February 2006) is much shorter, the shape of mass flux
profiles in Figure 5b is quite similar to their observations (Figure 4A, which is from Figure 13 of
Kumar et al. 2016 © American Meteorological Society. Used with permission).

1        Figure 5 shows the convective heating rate of shallow (Fig. 5A), congestus (Fig. 5B), and

deep convection (Fig. 5C). In the case of the shallow convection (Fig. 5A) the environment is
warmed in the lower levels and cooled at cloud tops. Temperature tendencies are derived using

$$\frac{\partial T}{\partial t} = \frac{1.}{c_p}\varrho[h(z)]m_{b(CU)} - \frac{L_v}{c_p}\varrho[q(z)]m_{b(CU)} \quad (23)$$

Here the $\varrho$ is the change of moist static energy ($h$) or water vapor ($q$) per unit mass, and $m_{b(CU)}$ is
the cloud base mass flux for deep, congestus, or shallow convection.

8        The shallow heating by shallow convection appears more active in the monsoon stage. The

congestus (Fig. 5B) and deep (Fig. 5C) convection cool the boundary layer mainly by downdrafts
and evaporation of rainfall, and also cool the troposphere by the evaporation of the detrained cloud
condensates at cloud tops. On 23 January 2006, the strong heating from lower troposphere to
500hP and 200hPa for congestus and deep convection, respectively, corresponds to the heavy
precipitation in Figure 3. Figure 6 shows the convective drying tendencies of shallow (Fig. 6A),
congestus (Fig. 6B) and deep convection (Fig. 6C). The entraining of low-level environmental
moist air into the convection plumes and raining out results in drying of low-level atmosphere,
while the detrained cloud water/ice at the cloud top leads to some cooling. The strongest drying
for deep convection on 23 January 2006 (Fig. 6C) from lower troposphere to 200hPa also
corresponds to the heavy precipitation in Figure 4.

19        The heating and drying features of the SCM simulation with the GF convection scheme are

further validated with diabatic heating source (Q1) and drying sink (Q2), which were defined by
Yanai et al. (1973), from sounding analysis. The averaged profiles from Q1 and Q2 derived from
constrained variational objective analysis observation (Xie et al. 2007) are shown in Figure 7A
and Figure 7C, while the SCM simulated Q1 and Q2 are given in Figure 7B and Figure 7D. The
shape of Q1 and Q2 in active/suppressed periods from simulation agrees with the observation very
well, but with stronger magnitude. The maximum of Q1 and Q2 between 350hPa and 550 hPa in
active monsoon period corresponds to the heavy precipitation in Figure 3. The Q1 and Q2 from
observation and simulation were mainly distributed at low levels in suppressed period, consisting
with the study from Xie et al. (2010).
**3.2) Evaluation of the Diurnal Cycle Closure**
Santos e Silva et al. (2009, 2012) discussed in detail the diurnal cycle of precipitation over the
Amazon Basin using the TRMM rainfall product (Huffman et al., 2007) and observational data

from an S-band polarimetric radar (S-POL) and rain gauges obtained in a field experiment during the wet season of 1999. Their analysis indicated that the peak in rainfall is usually late in the afternoon (between 17:00 and 21:00 UTC), despite existent variations associated with wind regimes. In addition, over the Amazon, a secondary convection activity is observed during the nocturnal period as reported by Yang et al. (2008) and Santos e Silva et al. (2012); in general, this is associated to squall lines propagation in the Amazon basin (Cohen et al., 1995; Alcantara et al., 2011). This bimodal pattern of convective activity can be identified with observational analysis of vertical profiles of moistening and heating (Schumacher et al., 2007).

Here we evaluate the GF scheme with the B2014 closure, applying it with the NASA GEOS GCM configured as a single column model (SCM). The GEOS with GF was run as a SCM from 24 January to 25 February 1999 using the initial condition and advective forcing from the TRMM_LBA field campaign data. The simulation started on 00Z 24 January 1999 with 1 month time integration. Model results were averaged in time to express the mean diurnal cycle. An initial glance at the three convection modes in the GF scheme is given by Figure 8, where the time averaged mass fluxes ($10^{-3}$ kg m$^{-2}$ s$^{-1}$) of each mode are introduced. The contour lines in black represent the vertical diffusivity coefficient for heat (m$^2$ s$^{-1}$), describing the diurnal development of the planetary boundary layer (PBL) over the Amazon forest. The PBL development seems to be well represented with a fast evolution in the first hours after the sunrise and stabilizing around noon with a realistic vertical depth between 1 and 1.5 km. Both shallow (Fig. 8A) and congestus (Fig. 8B) modes start few hours after sunrise with cloud base around the PBL height and cloud tops below ~ 700 and 550 hPa, respectively. Those two modes precede the deep convection (Fig. 8C) development during the late afternoon (local time is UTC – 4 hours) with cloud tops reaching 200 hPa.

Figure 9 shows the mean diurnal cycle of the net vertical mass flux (the sum of shallow, congestus and deep modes) as well as the total and convective precipitation. The chosen closures for the mass flux at cloud base were the BLQE for shallow and the adaptation of B2014 for congestus and deep modes, as described at the end of Section 2.3. For congestus, we only retained the first term of Equation 17; for deep, the simulations were performed without and with the second term of Equation 17. This allowed us to evaluate its role on defining the phase of the diurnal march of the precipitation.

Figure 9A shows the model results without applying the diurnal cycle closure (i.e. retaining only
the first term of Eq. 17) for deep convection. In this case, the three convective modes coexist,
triggered just a few hours after the sunrise (∼11 UTC), with the deep convection occurring too
early and producing a maximum precipitation of about 15 UTC (∼11 Local Time). Conversely,
we observed a clear separation between the convective modes when applying the full equation of
the diurnal cycle closure (Fig. 9B), reducing the amount of potential instability available for the
deep convection. In this case, there is a delay of the precipitation from the deep penetrative
convection with the maximum rate taking place between 18 and 21 UTC, more consistent with
observations of the diurnal cycle over the Amazon region.
Figure 10 introduces the grid-scale vertical moistening (on the left) and heating (on the right)
tendencies associated with the three convective modes for the simulations with- out and with the
diurnal cycle closure. The net effect (moistening minus drying) of the three convective modes, not
including the diurnal cycle closure for the deep mode, appears in the Figure 10A. As the three
modes co-exist most of the time and as the drying associated with the deep precipitating plumes
dominates, water vapor is drained from the troposphere, with a shallow lower-level layer of
moistening associated with the precipitation evaporation driven by the downdrafts. However, by
including the full formulation of the diurnal cycle closure (Fig. 10B), a much smoother transition
is simulated with a late morning and early afternoon low/mid-tropospheric moistening by shallow
and congestus convection, followed by a late afternoon and early evening tropospheric drying by
the rainfall from the deep cumulus. Associated with the delay of precipitation, the peak of
downdrafts occurrence is correspondingly displaced. On the right, Figure 10C and Figure 10D
introduce the results for the heating tendencies. A similar discussion applies to these tendencies,
with the peak of the atmospheric heating delayed by a few hours, when the diurnal cycle closure
is applied (Fig. 10D). Note, the warming from the congestus plumes somewhat offsets the low-
troposphere cooling associated with the shallow plumes.
**3.3 Global Scale 3-dimensional Modeling**
A global scale evaluation of the diurnal cycle closure is shown in this section applying GF within
the NASA GEOS GCM model (Molod  et al., 2015). The GEOS GCM was configured with c360
spatial resolution (∼ 25km) and was run in free forecast mode for all of January 2016. Each forecast
day covered a 120-h time integration, with output available every hour. Atmospheric initial

conditions were provided by the Modern–Era Retrospective Analysis for Research and Applications, Version 2 (MERRA–2, Gelaro et al., 2017). The simulations applied the FV3 non-hydrostatic dynamical core on a cubed-sphere grid (Putman and Lin, 2007). Resolved grid-scale cloud microphysics applies a single-moment formulation for rain, liquid and ice condensates (Bacmeister et al., 2006). The longwave radiative processes are represented following Chou and Suarez (1994), and the shortwave radiative processes are from Chou and Suarez (1999). The turbulence parameterization is a non-local scheme primarily based on Lock et al. (2000), acting together with the local first order scheme of Louis and Geleyn (1982). The sea surface temperature is prescribed following Reynolds et al. (2002).

We first demonstrate the impact of the boundary layer production on the cloud work function (CWF) available for the deep convection overturning. Figure 11 shows the monthly mean of the diurnal variation of the three quantities given by Equations 10, 11 and 12. The figure represents the monthly mean (January 2016) of the diurnal variation of the total cloud work function, boundary layer production, and the available cloud work function all areal-averaged over the Amazon Basin.

The total CWF tightly follows the surface fluxes as the air parcels that form the convective updrafts originate close to the surface in the PBL. The boundary layer production presents similar behavior, peaking at noon and developing negative values during the nights. The combination of the two terms following the Equation 17 defines the available CWF for convection overturning. Note, a negative range of the available CWF in the early mornings to approximately noon, prevents the model from developing convective precipitation in that period and shifting the maximum CWF to late afternoon, much closer to the observed diurnal cycle of precipitation over the Amazon region.

A global perspective of these three quantities is shown in Figure 12. As before, the curves represent the monthly mean (January 2016) of the diurnal variation of the total cloud work function, boundary layer production, and the available cloud work function. Here the averaged areal corresponds to the global domain (Fig. 12A), only the land regions (Fig. 12B) and only the oceans (Fig. 12C). Over oceans, the boundary layer production is small in comparison with the total CWF, and does not do much; instead, over land (Fig. 12B), it is comparable in magnitude with the total CWF, pushing the available CWF to peaks closer to the late afternoons and early evenings. On global average (Fig. 12A), the boundary layer production still plays a substantial role with a clear effect in the timing of the maximum available CWF.

A perspective of the precipitation simulation with GEOS-5 GCM with the GF scheme and the impact of the diurnal cycle closure is provided by Figure 13. Here, the January 2016 average of the diurnal cycle of the precipitation (left column) and the July 2015 (right column) are depicted. Figure 13 A and D show the rainfall estimation by the TRMM Multi-satellite Precipitation Analysis (TMPA version 3B42, Huffman et al., 2007). Also, the simulated by the GEOS-GF, including (at middle, Fig. 13 C and E) or not (lower panels, Fig 13. D and F) the diurnal cycle closure are depicted. The precipitation fields were averaged over the latitudes between 40 S and 40 N taking into account only the land regions. The vertical axis represents the local time.

The TRMM estimation evidences two peaks of precipitation rate: a nocturnal one around 3 AM over oceans (not shown) and another one in late afternoon (3 to 6 PM) over land. A significant gap of rainfall in the mornings is also seen in both months. We found a somewhat overestimation of the precipitation in comparison with the estimates produced by the TRMM retrieval technique (Fig. 13 A and D). However, the simulations that applies the diurnal cycle closure (Fig. 13 C and F) are superior regarding the phase in comparison with the simulation which applies the total CWF (Fig. 13 B and E) for the closure. As shown in Figure 13C and 13F, the diurnal cycle closure adapted from B2014 used in these simulations show a much better representation of the morning to early afternoon gap of the precipitation, which peaks much closer to the time of TRMM retrieval. In particular, model improvements are noticeable over the Amazon region (denoted by "South America"). Similar improvements are also evident over Africa and Australia.

For more detailed analysis of the diurnal cycle of the precipitation we use higher spatial and temporal resolution retrievals from the Global Precipitation Measurement (GPM) with the Integrated Multi-satellitE Retrievals for GPM (IMERG, version 6, *Huffman et al.,* 2019). The IMERG has 0.1-degree spatial and ½ -hour temporal resolutions. Also, we adopt the technique of calculating the diurnal harmonics using a Fourier transform and focus on the phase and amplitude of the first harmonic. The GPM IMERG retrievals were first interpolated to the GEOS-5 grid spatial resolution (~ 25 km) e temporal accumulation (1-hour). Figure 14 shows the mean precipitation, and the mean amplitude and phase of the first harmonic over the Amazon Basin. The diurnal phase was shifted to the local solar time (LST) and 12 LST is associated with the time of maximum insolation in a cloud free sky condition. The IMERG mean precipitation (Fig. 14A) shows the typical summer pattern over the Amazon Basin with the maximum accumulated precipitation occurring South of Equator following the annual southward shift of the Inter Tropical

Convergence Zone (ITCZ). The domain average precipitation estimated by IMERG was 5.62 mm day$^{-1}$. The correspondent field as simulated by the GEOS-5 is shown in Fig 14D and G without (DC OFF) and with (DC ON) the adaptation of B2014 diurnal cycle closure, respectively. Both simulations show a very similar pattern, and they are also reasonably comparable with the IMERG in the inner part of the continent. However, the simulations suffer from spurious precipitation along the Andes mountains triggered by numerical noise associated with the steep terrain and the use of a sigma-type vertical coordinate. The simulated domain average precipitation was 6.69 (6.59) mm day$^{-1}$ for the case DC OFF (ON), roughly 18% larger than IMERG. It seems plausible that precipitation excess is mostly associated with the spurious generation along the steep terrain. The central column of Figure 14 shows the January 2016 mean amplitude of IMERG (panel B) and model simulations (panels E and H). The domain average amplitude corresponds to 61, 51 and 62% of the precipitation of IMERG, model DC OFF and DC ON, respectively. The column at right of Figure 14 shows the diurnal phase of the three data sets. Following Kousky (1980) the maximum precipitation which forms just inland along the coast at late afternoon is associated with the development of the sea breeze front. With the sea breeze further inland penetration, other maximum occurs during the nighttime with the convergence formed with the onshore flow. Both features are present in the simulations (Fig. 14 F and I), but the case DC ON better simulates the timing, being closer to the IMERG. As for the Amazon Basin interior, the IMERG shows a nighttime maximum associated with the squall lines that form along the northern coast of Brazil and propagate for long distances across the basin (Alcântara et al., 2011). Both simulations were not able to capture the propagation of these convective lines. However, it is clear that the case DC OFF (Fig. 14F) simulates a maximum of amplitude too early, between 10 and 14 LT, whereas the case with the diurnal cycle ON (Fig. 14I) is closer to the timing of the IMERG (Fig. 14C), with the peaks occurring between 14 and 18 LT.

Correspondent analysis over a portion of the Pacific Ocean is discussed as follows. Figure 15 shows the three datasets for the tropical Pacific Ocean for January 2016. The domain average precipitation estimated by IMERG was 4.53 mm day$^{-1}$, whereas GEOS-5 with DC OFF and DC ON simulated ~ 4.21 mm day$^{-1}$ in both configurations. For the amplitudes, the amounts were 2.16, 1.47, and 1.45 mm day$^{-1}$, respectively. The column at the left of Figure 15 shows that the spatial distribution of the precipitation simulated by GEOS-5 (Panels D and G) remarkably resembles the IMERG retrieval (Panel A), although the domain average precipitations are underestimated by

about ~ 10%. The former discussion also applies to the amplitudes, as shown in the central column
of Figure 15. For the phase, most of the precipitation peaks occur through the nighttime (Panel C),
and the simulations with GEOS-5 have a similar pattern. The fact that both simulations are nearly
the same in terms of the precipitation amounts and its diurnal cycle over the ocean is explained by
Figure 12C.
The diurnal cycle of precipitation of the north equatorial portion of Africa for July 2015 is
discussed based on the results shown in Figure 16. The domain average precipitation (amplitude)
correspondent is 2.51 (2.12), 2.79 (1.45), and 2.8 (1.8) mm day$^{-1}$ for the panels A (B), D (E), and
G (H), respectively. Note that the simulated mean precipitations are about 11% larger than the
IMERG estimation. For the diurnal phase (Fig. 16C), the IMERG retrieval shows a mix of late-
afternoons (16 – 20 LT) and nighttime (00 – 04 LT) maximum amplitudes. As before, the
simulations show contrasting results for the timing of precipitation. Not accounting for the diurnal
cycle closure, results with the precipitation peaks occurring too early (mostly 10 – 14 LT, Fig.
16F), whereas with that closure, those peaks take place mainly after 14 - 16 LT (Fig. 16I).
Figure 17 displays the results for July 2015 over the contiguous United States and part of the
neighbor's countries. The domain average precipitation (amplitude) correspondent is 2.60 (2.37),
2.52 (1.59), and 2.42 (1.8) mm day$^{-1}$ for panels A (B), D (E), and G (H), respectively. Model
simulations underestimate the mean precipitation by about 5 – 10%. As for the other regions, the
model's monthly mean spatial distribution of the precipitation looks realistic. Still, it
underestimates the southern region's amount and overestimates over the east part of Gulf of
California. According with IMERG, the peaks of precipitation occur in the late afternoon over the
southeast and central-west part of the region. And in the nighttime over the central-east part of the
domain (Fig. 17C). Over the central part of the U.S., both simulations did not capture the nighttime
precipitation well. However, the simulation DC ON (Fig. 17I) seems to be closer to IMERG over
the central-west portion.

## 4 Conclusions

We describe a set of new features recently implemented in the GF convection parameterization. The main new aspects are as follows:

- The unimodal approach has been replaced by a trimodal formulation representing the three modes: shallow, congestus, and deep convection. Each mode has a distinct initial gross entrainment and a set of closure formulations for the mass flux at the cloud base.

- The normalized mass flux profiles are now prescribed following a continuous and smooth probability density function. From the cloud base, cloud top, and a free parameter, which shapes the PDF, the normalized mass flux profile, the entrainment and detrainment rates are determined. Together with the mass flux at the cloud base defined by the selected closure, they also determine, e.g., the vertical drying and heating tendencies associated with the sub-grid-scale convection. Using a PDF to describe the statistical average of a characteristic convection type means that the PDF may in fact represent several plumes in the grid box.Additionally, this approach may be used to implement stochasticism with temporal and spatial correlations and memory dependence that lead to significant changes in the vertical distribution of heating and drying without disturbing mass conservation. Future work will address this possibility. Finally, the use of the PDFs may help fine-tuning the model skill by removing water vapor and temperature biases.

- An optional closure for non-equilibrium convection updated from Bechtold et al. (2014) is available. This closure has shown a significant gain of the GF scheme's ability in NASA GEOS GCM in representing the diurnal cycle of convection over land, with potential beneficial impacts also in data assimilation and tracer transport.

We understand that the previous GF scheme's features with the new ones described in this paper, further extends the capabilities of this convection parameterization to be applied in a wide range of spatial scales and environmental problems.

**Code availability**

The GF convection scheme within the Global Model Test Bed (GMTB) Single Column Model is available at the GMD-paper branch of https://github.com/GF-GMD/gmtb-scm. Public access to

the NASA GEOS GCM source code is available at github.com/GEOS- ESM/GEOSgcm on tag Jason-3.0. The authors are available for recommendations of the applying the several options present in the GF scheme, as well as for instructions for its implementation in other modeling systems.

**Competing interests**

The authors declare that they have no conflict of interest.

**Author contribution**

SRF and GAG developed the model code and performed the simulations. HL conduced the simulations and produced the results shown in Section 3.1. All authors prepared the manuscript.

**Acknowledgements**

The first author acknowledges the support of NASA/GFSC - USRA/GESTAR grant # NNG11HP16A. This work was also supported by the NASA Modeling, Analysis and Prediction (MAP) program. Computing was provided by the NASA Center for Climate Simulation (NCCS).

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

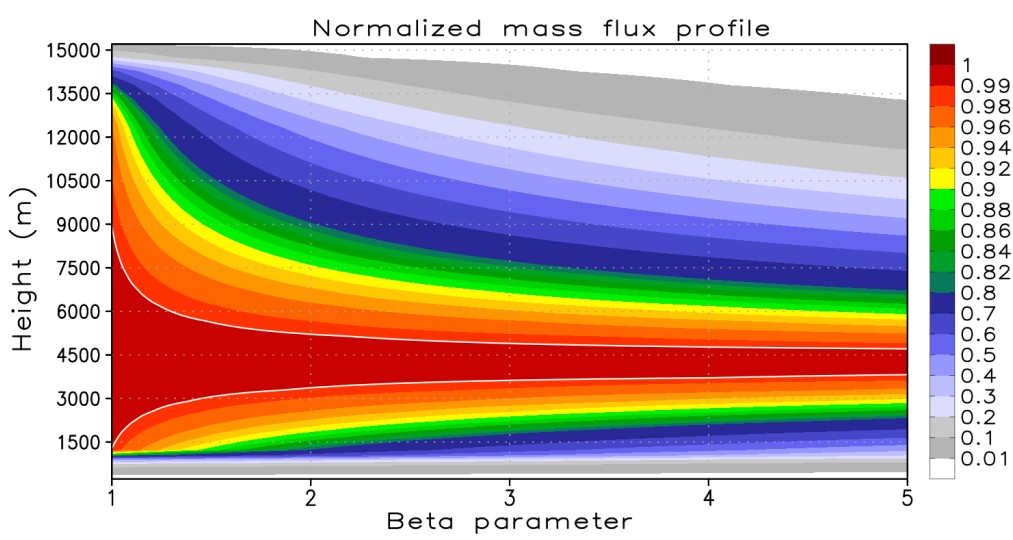

Figure 1. The universe of solutions for the normalized updraft mass flux profile ($Z_u$) for a case of
the cloud base resides at 1.2 km height, the height of maximum $Z_u$ is 4.3 km, and the cloud top is
at 15.1 km height. The horizontal axis denotes the range of variation of the beta parameter. The
white contour lines delimit the solution domain where $Z_u \in [0.99, 1.]$.

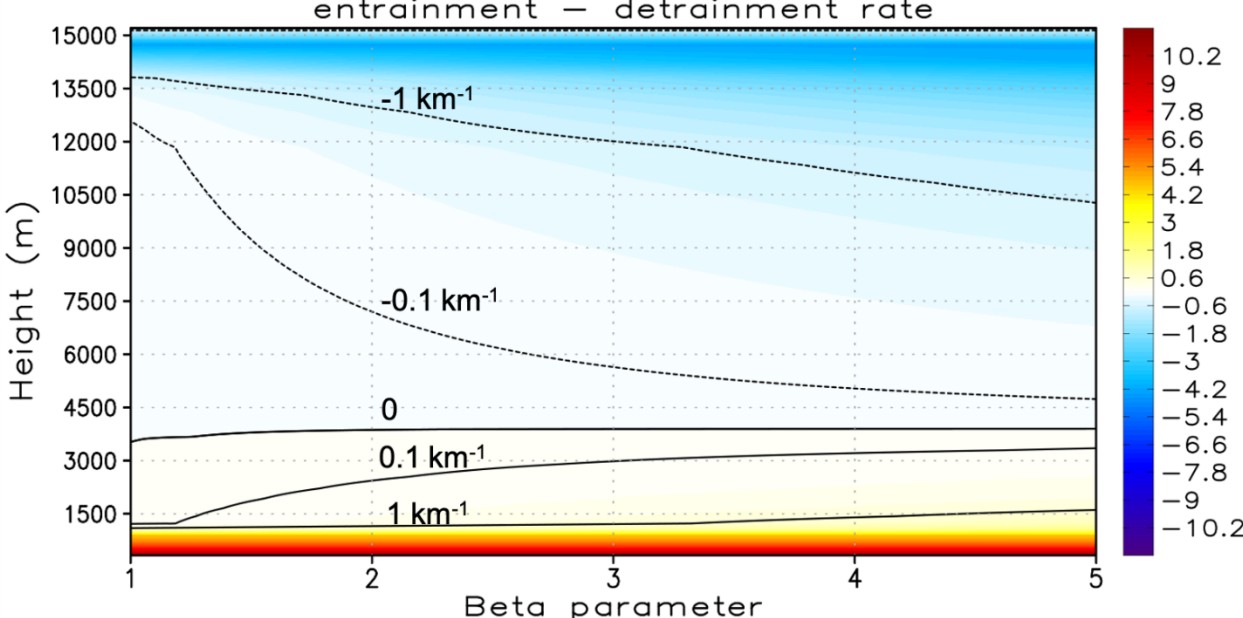

Figure 2. The universe of solutions for the effective net mass exchange rate (entrainment –
detrainment, $km^{-1}$) for the case shown in Figure 1. The black contour lines demark the transition
from mostly entraining to mostly detraining plumes.

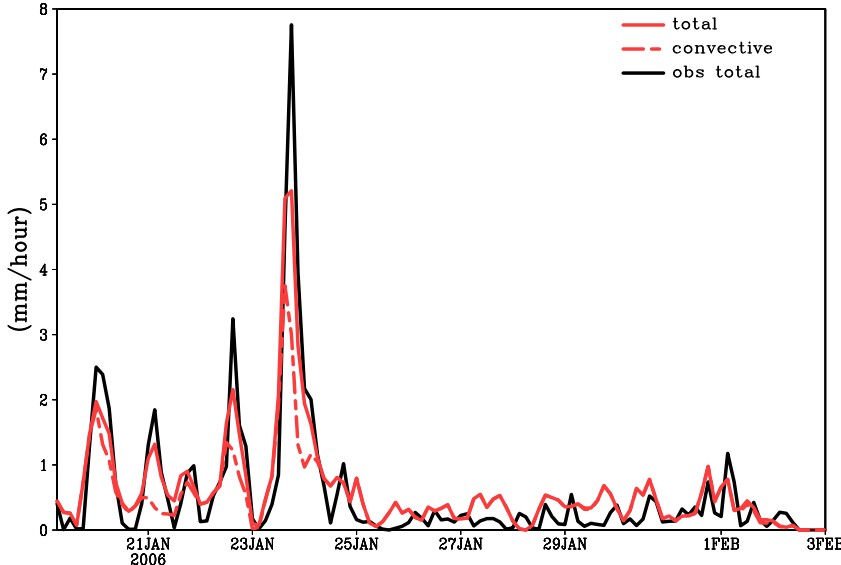

Figure 3. Total (red solid), convective (red dashed) and observed total precipitation rates
(mm/hour) with GF scheme using the TWP-ICE soundings.

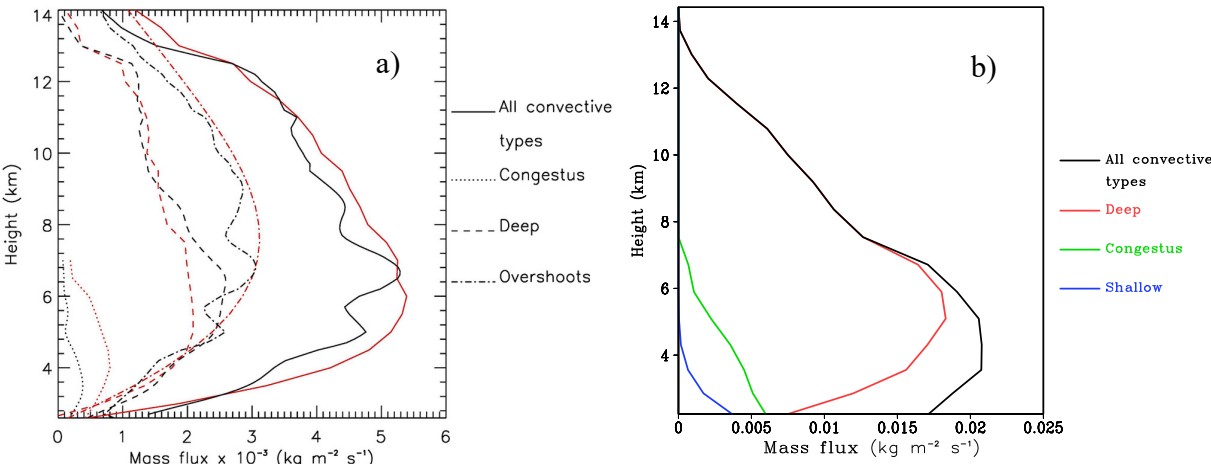

Figure 4. On the left, two season's mean mass flux associated with all cumulus clouds (solid curves), congestus (dotted), deep (dashed), and overshooting convection (dotted-dashed) using wind-profiler (black) and CPOL-based (red) measurements taken at the profiler site (Kumar, V.V., et al.: The Estimation of Convective Mass Flux from Radar Reflectivities. J. Appl. Met. Clim., 55, 1239–1257, 2016. © American Meteorological Society. Used with permission). (b) On the right, the TWP-ICE mean mass flux (kg m$^2$ s$^{-1}$) profiles from all cumulus clouds (in black), shallow (in blue), congestus (in green), and deep convection (in red) with GF SCM simulation.

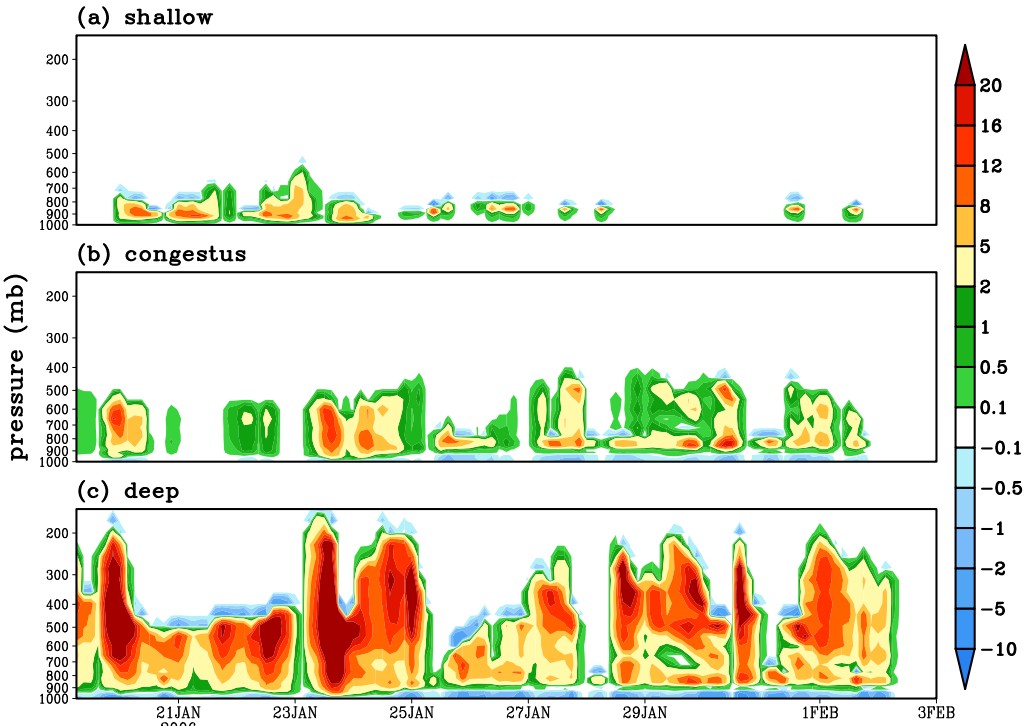

Figure 5. Convective heating tendencies (K day⁻¹) of (a) shallow, (b) congestus, and (c) deep
convection with GF scheme using the TWP-ICE soundings.

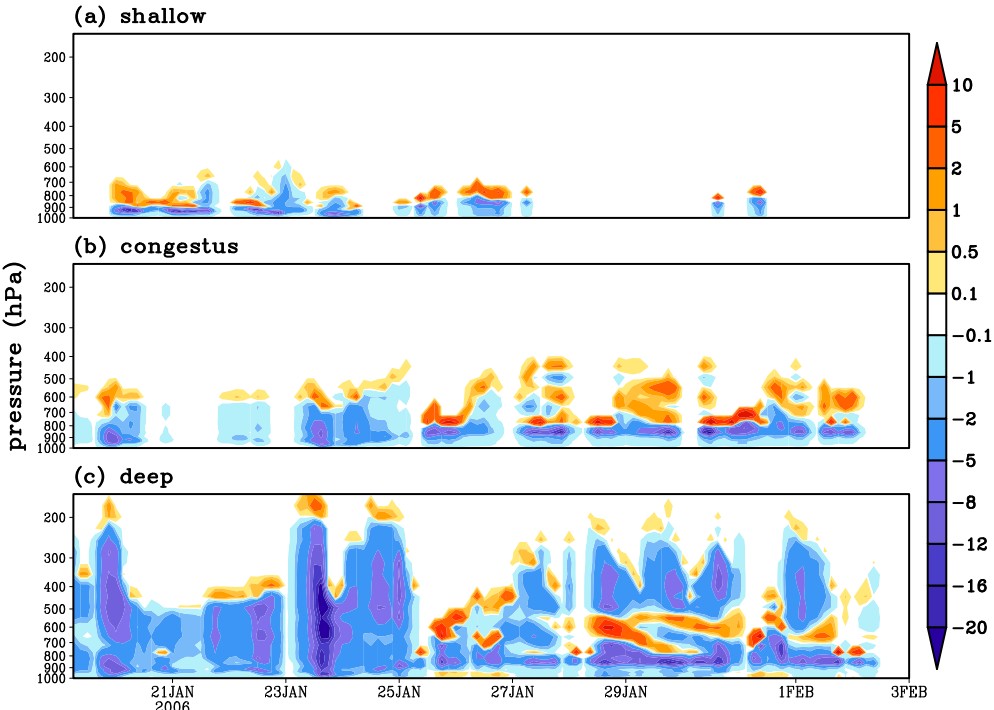

Figure 6. Convective drying tendencies (g kg⁻¹ day⁻¹) of (a) shallow, (b) congestus, and (c) deep
convection with GF scheme using the TWP-ICE soundings.

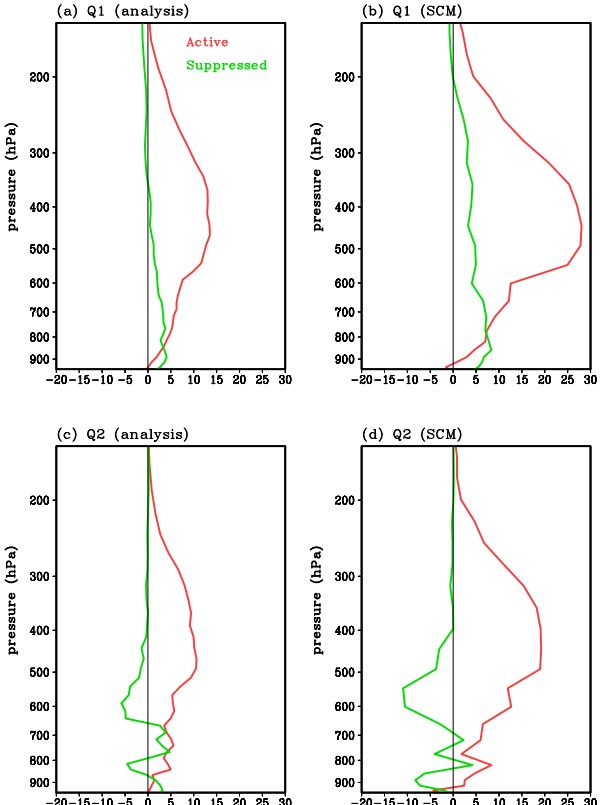

Figure 7. The diabatic heating source (Q1, K day$^{-1}$) profiles from (a) sounding analysis, (b) SCM
simulation, and diabatic drying sink (Q2, K day$^{-1}$) from (c) sounding analysis, (d) SCM simulation.
The active monsoon period is in red, and the depressed period is in green.

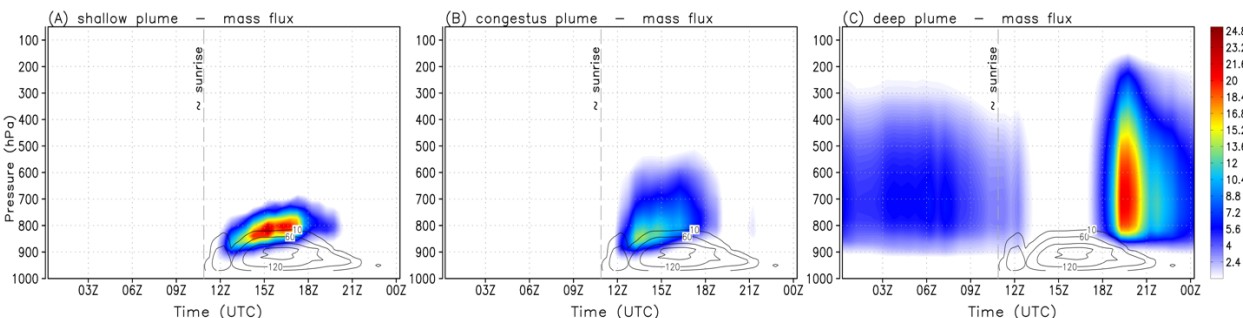

Figure 8. The diurnal cycle of the three convective modes as represented by the GF convection
parameterization in a single column model (SCM) experiment with the GEOS-5 modeling system.
The black contours represent the vertical diffusivity coefficient for heat (m$^2$ s$^{-1}$). The color bar
represents the updraft mass flux expressed in 10$^{-3}$ kg m$^{-2}$ s$^{-1}$.

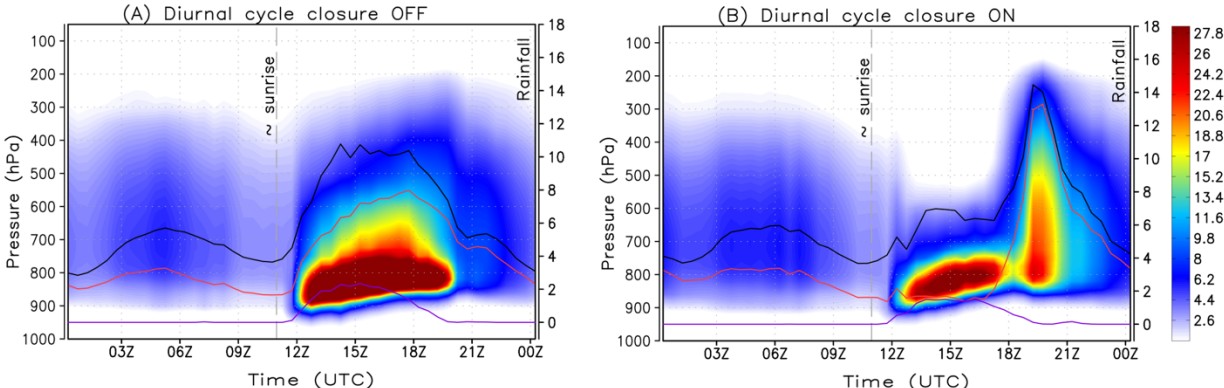

Figure 9. Time average of the diurnal cycle of the total vertical mass flux of the three convective modes: shallow, congestus, and deep ($10^{-3}$ kg m$^2$ s$^{-1}$). The rainfall is depicted by graphic lines: black, red and purple represent the total precipitation, and the convective part from deep and congestus plumes, respectively. The scale for rainfall appears on the right vertical axis (mm day$^{-1}$). Panel A (B) represents the results without (with) the diurnal cycle closure.

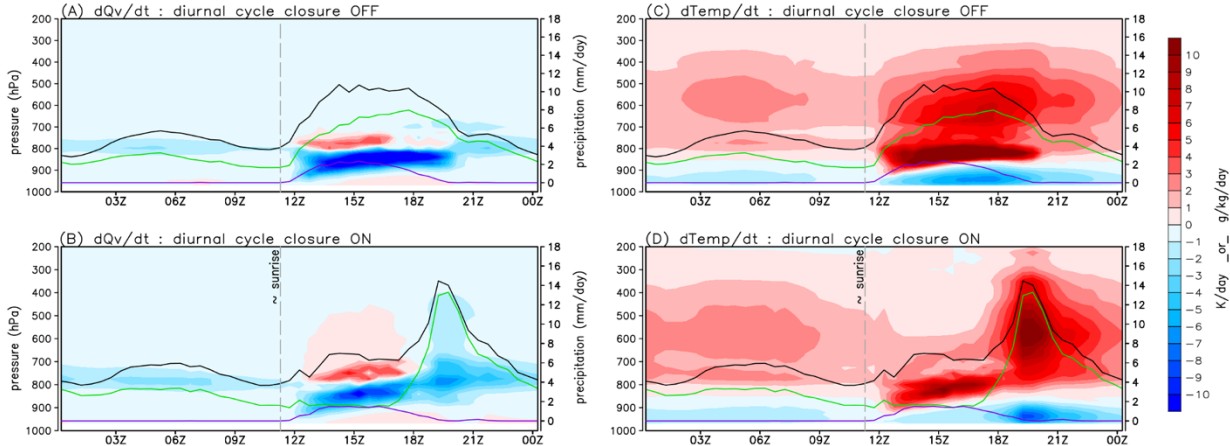

Figure 10. Time average of the diurnal cycle of the grid-scale vertical moistening (left) and heating (right) tendencies associated with the three convective modes (shaded colors) and precipitation (contour: red dash, green solid and purple dash represents the total precipitation, and the convective precipitation from deep and congestus plumes, respectively). The upper (bottom) panels show results without (with) the diurnal cycle closure.

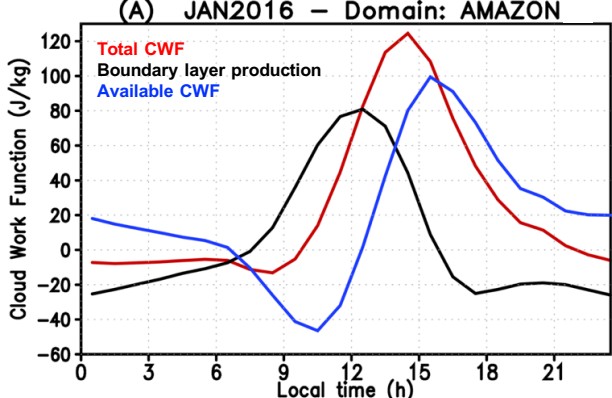

Figure 11. The monthly mean (January 2016) of the diurnal variation of the total cloud work
function (red color), boundary layer production (black) and the available cloud work function
(blue). The curves also represent the areal average over the Amazon region.

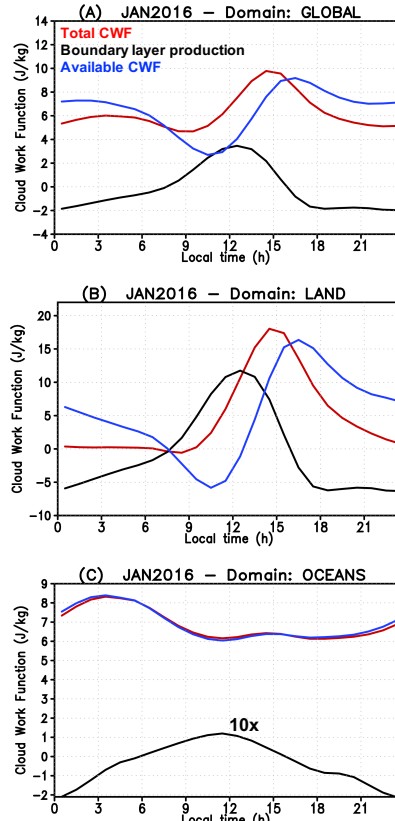

Figure 12. The monthly mean (January 2016) of the diurnal variation of the total cloud work
function (red color), boundary layer production (black) and the available cloud work function
(blue). The curves also represent the areal average over (A) the entire globe, (B) the land regions,
and (C) the oceans. In panel (C) the boundary layer production is multiplied by 10 for clarity.

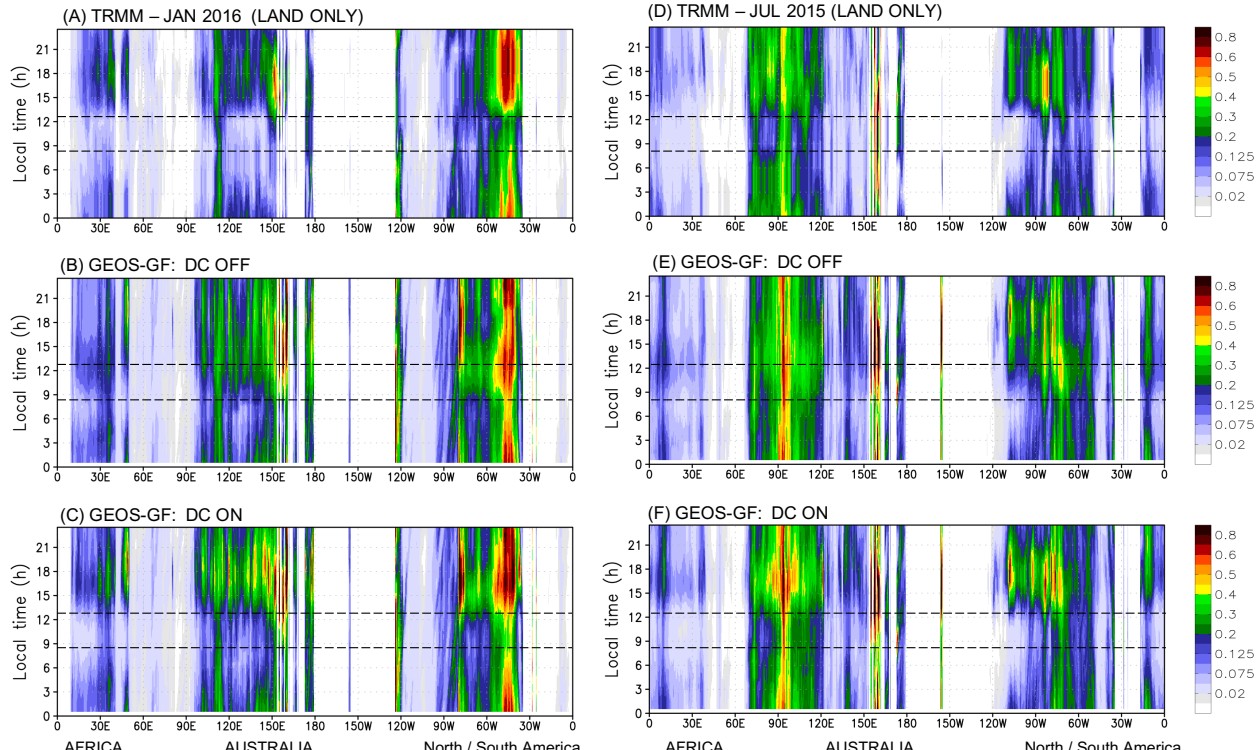

Figure 13. Global Hovmöller Diagram (average over latitudes 40S to 40N) of the diurnal cycle of
precipitation (mm h$^{-1}$) from remote sensing-derived observation (TRMM, upper panels) and
NASA GEOS GCM applying the GF scheme without the diurnal cycle closure (middle panels, DC
OFF) and with (lower panels, DC ON). The results account for precipitation only over land regions
and are monthly means for January 2016 (left column) and July 2015 (right column), respectively.

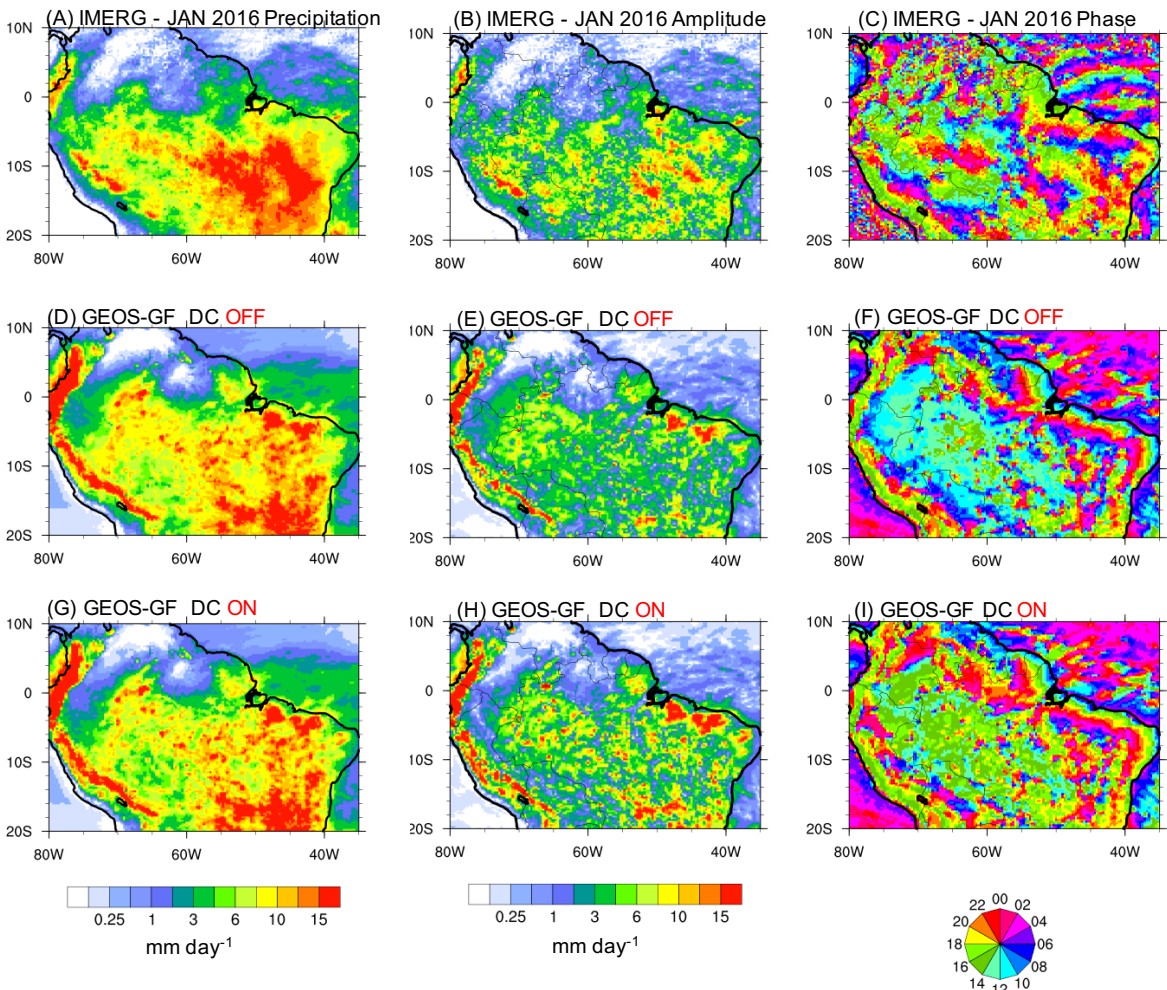

Figure 14. The January 2016 monthly mean precipitation, amplitude, and phase of the diurnal
harmonic over the Amazon Basin. The top panels show the quantities of the GPM IMERG
retrieval. In the middle and lower rolls, panels show model simulations with the diurnal cycle
closure turned OFF and ON, respectively.

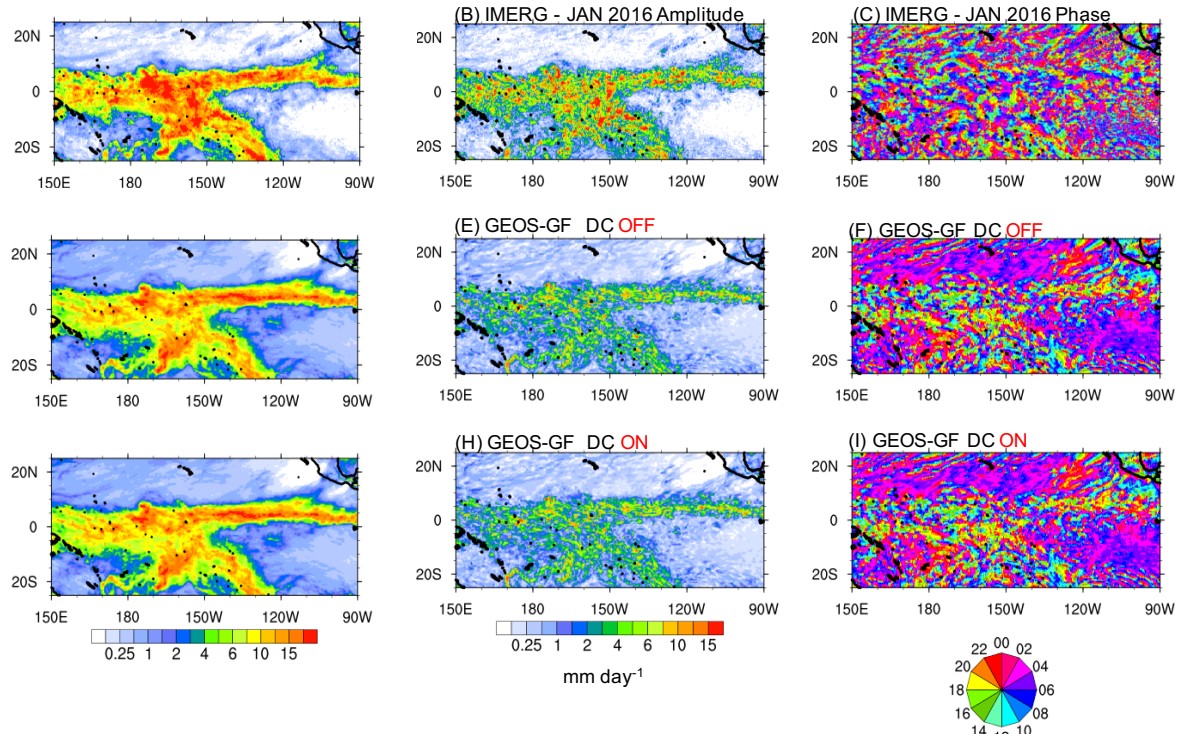

Figure 15. The January 2016 monthly mean precipitation, amplitude, and phase of the diurnal
harmonic over the Tropical Pacific Ocean. The top panels show the quantities of the GPM IMERG
retrieval. In the middle and lower rolls, panels show model simulations with the diurnal cycle
closure turned OFF and ON, respectively.

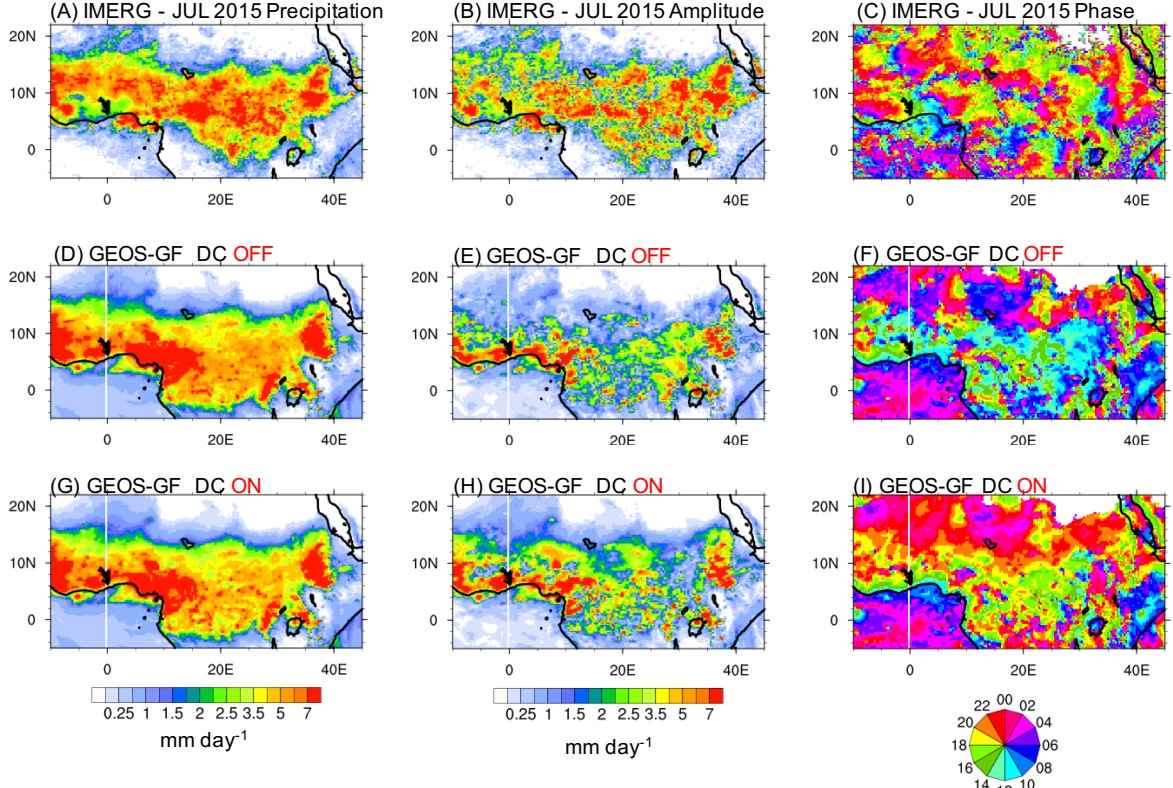

Figure 16. The July 2015 monthly mean precipitation, amplitude, and phase of the diurnal
harmonic over a portion of the Equatorial Africa. The top panels show the quantities of the GPM
IMERG retrieval. In the middle and lower rolls, panels show model simulations with the diurnal
cycle closure turned OFF and ON, respectively.

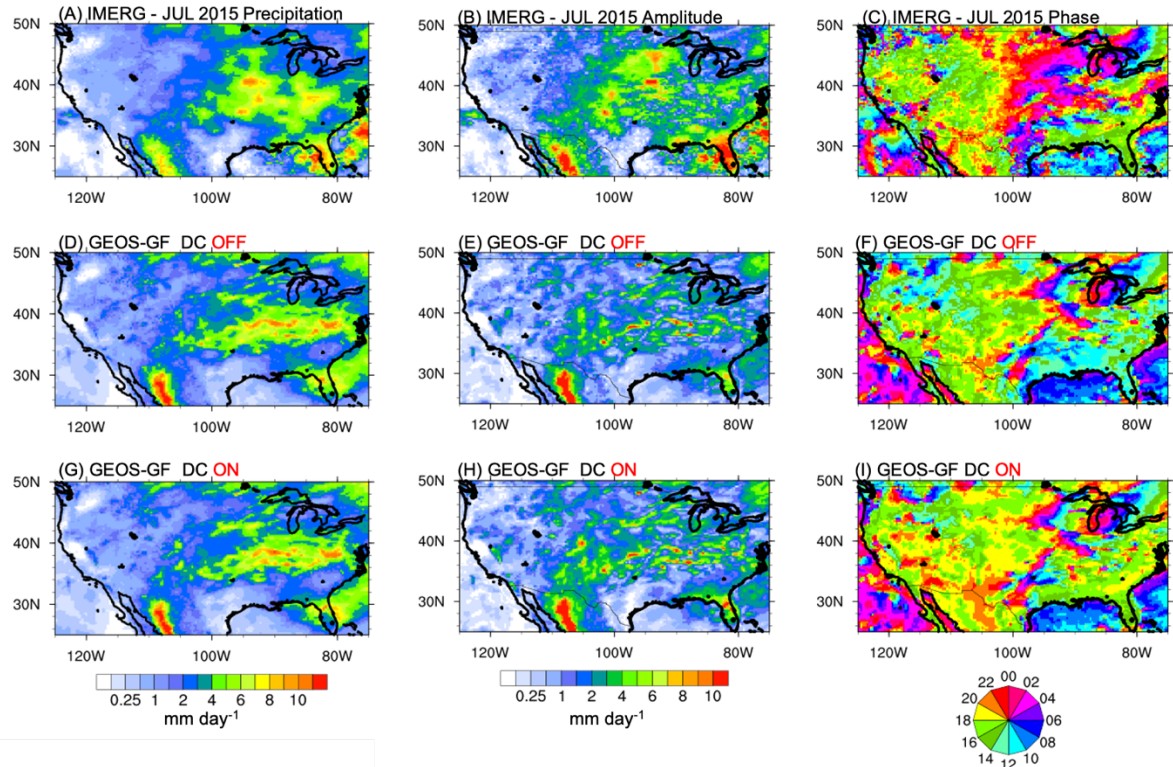

Figure 17. The July 2015 monthly mean precipitation, amplitude, and phase of the diurnal
harmonic over contiguous United States and part of the neighbor's countries. The top panels show
the quantities of the GPM IMERG retrieval. In the middle and lower rolls, panels show model
simulations with the diurnal cycle closure turned OFF and ON, respectively.

