# Peer review of "extensions, and applications"

_Geoscientific Model Development, 2020_

## Short Comment (SC1) · 24 Apr 2020

Dear authors,

in my role as Executive editor of GMD, I would like to bring to your attention our Editorial version 1.2:

https://www.geosci-model-dev.net/12/2215/2019/

This highlights some requirements of papers published in GMD, which is also available on the GMD website in the 'Manuscript Types' section: http://www.geoscientific-model-development.net/submission/manuscript_types.html

In particular, please note that for your paper, the following requirement has not been met in the Discussions paper:

- "The main paper must give the model name and version number (or other unique identifier) in the title."

- Code must be published on a persistent public archive with a unique identifier for the exact model version described in the paper or uploaded to the supplement, unless this is impossible for reasons beyond the control of authors. All papers must include a section, at the end of the paper, entitled "Code availability". Here, either instructions for obtaining the code, or the reasons why the code is not available should be clearly stated. It is preferred for the code to be uploaded as a supplement or to be made available at a data repository with an associated DOI (digital object identifier) for the exact model version described in the paper. Alternatively, for established models, there may be an existing means of accessing the code through a particular system. In this case, there must exist a means of permanently accessing the precise model version described in the paper. In some cases, authors may prefer to put models on their own website, or to act as a point of contact for obtaining the code. Given the impermanence of websites and email addresses, this is not encouraged, and authors should consider improving the availability with a more permanent arrangement. Making code available through personal websites or via email contact to the authors is not sufficient. After the paper is accepted the model archive should be updated to include a link to the GMD paper.

Please provide the version number of the GF convection scheme in the title of your revised manuscript.

Additionally, please note, that GMD requires the authors to provide a persistent access to the exact version of the source code used for the model version presented in the paper. As explained in https://www.geoscientific-model-development.net/about/manuscript_types.html the preferred reference to this release is through the use of a DOI which then can be cited in the paper. For projects in GitHub a DOI for a released code version can easily be created using Zenodo, see https://guides.github.com/activities/citable-code/ for details.

Yours, Astrid Kerkweg
* * *

---

## Referee Comment (RC1) · Anonymous Referee #1 · 26 Apr 2020

pre-script: This is my first interaction with a Discussion and Comments journal, so please forgive any cultural blunders. I have mostly considered this a low-quality approach, and as a reader I mostly ignore these dubious new forms of "publication". But I appreciate the experiment, and was curious about this topic, so I agreed to it. I hope revision will be as incentivized in this format as it would be in a real (traditional) journal process.

("general comments")

This manuscript surveys some aspects of the latest version of the GF convection scheme, whose code is offered in an open source repository. I suppose the world

is fortunate to have a narrative record from busy hard-working developers about what they have been doing. But I found it fairly unclear, despite my appetite for understanding it, and in places a bit congratulatory or justifying where the same column inches could be spent on more information. Major revision is needed if this is meant to reach the standards of a formal "scientific publication" in a traditional sense.

Since a lot of prior knowledge of the subject is assumed, and since the text has little pretense to a physical rationale for the algorithms being described, the paper could be very short and to the point: what is the informatic mapping from inputs and assumptions, to internal variables, to outputs? Unfortunately, a careful read of this manuscript is required to fish out many of these mappings and assumptions, and then many details are left ambiguous, for instance about what kinds of choices a user must make (and the authors DID make in the examples presented).

In short, it could use a tightening-up in style, and a completeness check for crucial details. Here is an indented bulleted list of the inputs, internals, and outputs as I see them. The formatting system seems to mess them up, I find, but this is what I offer as an unpaid volunteer; the reader can guess the hierarchy.

Inputs:

- T and q and wind and tracer profiles

- the Tv tendency averaged over the PBL from a separate BL scheme

- an aerosol-related input to autoconversion efficiency in the height domain

Assumed parameters:

- what level to start entraining parcels/plumes from, never explained

- 3 "initial" entrainment rates {2, 0.9, 0.1} /km , buried in different places

- something about the undetermined 4th parameter of the Beta profile curve

[Figure]

- closure choices for shallow convection - for 3 different approaches; and rules for how to select or combine them

- closure choices for middle and deep convection -assumed timescale for convection to damp the work function - the strength of a diurnal delay trick, misleadingly cast as a timescale tau_b

- temperatures for freezing and melting (melting should not begin at -3C, should it?)

Intermediate variables:

- 4 parameters of the parametric steady-state mass flux profiles for each of the 3 kinds of updraft plumes - 2 params: how to map Beta's [0,1] domain into levels, z, log-p, or p (various figures use all four)

- 2 params: shape of the profile - cast as the altitude of the maxiumum? - cast as the "mean cloud base"?  - inverted from the closure somehow, along with a detrainment profile? - is an entrainment profile different from constant also backed out?

- All of the above for downdrafts too? downdrafts are mentioned, but never specified

- Finite area fraction profiles for minor "scale aware" adjustments of eddy flux from MF?

- Special assumptions for momentum vs. other tracers? (reflecting "pressure effects")

Outputs:

- profiles of tendencies of state variables (or the final result of successively updated profiles?)

- surface rainfall rate

Unfortunately, the manuscript is not clear at this level.  It should be, since that seems to be its one job.

("specific comments")

Allowing for its very high-level narrative intent, the paper still needs major clarifications.

At the outset, the gross characterizations of the GF scheme's type and intent should be given. It is a steady-state updraft plume model, evidently with no downdrafts - oh wait, page 5 line 19 mentions them, presumably as inverted saturated plumes of the same kind? Precipitation removal is not mentioned, except to say that it can be made "aerosol aware" via an autoconversion of some sort, and can optionally be disabled entirely for the plume with 2/km initial entrainment rate. Momentum flux is mentioned in the abstract, but never in the paper, so presumably all quantities are simply transported identically, based on plume mass flux, is that correct? Or is there some pressure treatment for momentum? How does the scheme define its cloud base, for one or all plumes (LCL of surface air?) and its parcel properties (mixture up to that LCL, or does it entrain its way through the subcloud layer?) How about cloud top - is it the highest or lowest level of neutral buoyancy for the initial entraining-only instability-probing parcel calculation with its specified entrainment rate? (2, 0.9, and 0.1 /km; these three numbers are buried in the text).

The 3 plumes are said to be successive in time; does this mean the scheme updates the state profile within a call? Merely for its own internal accounting, such that it returns only one tendency profile to the mother model, or is this entangled inseparably with a model's time-marching scheme? Must the host model use the same conserved variables? (presumably with some fixed constant reference values for Cp, Lv, etc.?)

Once the cloud base and top are determined, these are somehow (it is very ambiguous) used to fit a Beta Distribution in the height (or level? or log-pressure? or pressure? Figures include all four) domain, as the mass flux (MF) profile. Presumably that defines the eddy flux of all scalars and the condensation rate? Although lines 1-7 of Page 6 make it sound like the heating and moistening rates can ALSO be specified, or that they TOO can be made smooth because MF is, that is not how tendencies work out for a smoothly varying mass flux through a sharp inversion. Beta is a two-parameter curve family on [0,1] so we are left to guess how THREE of Beta's FOUR parameters are set

by (1) cloudbase cb, perhaps the LCL of some undefined parcel origination level?, and (2) one of its neutral buoyancy levels?, and (3a) perhaps something that is different from cb called "average cloud base" (line 21 of page 5) or (3b) "the average height of mass flux maximum" (line 1 of page 6)? Very unclear. Read the code, I suppose. Beta's FOUR parameters are: the mapping of [0,1] onto height, and Beta's two parameters, leaving only one parameter "free", to be defined by ?the closure? From Fig. 3 (why are there 3 curves?), it appears that the [0,1] domain of Beta is not the same as some clipping or masking at cb and ct. From the mass flux profile, entrainment and detrainment are backed-out, although how TWO profiles are backed out of ONE is unclear. Is the buoyancy of the new entraining-detraining plume used to revisit the initial cloud base and top, or are these retained once fixed by the constant-entrainment-rate parcel buoyancy?

Quite unclear, all of this.

Freezing and melting are now accounted for, which seems fine and necessary, but for some reason melting begins at -3C, which makes no physical sense to me.

("technical corrections": typing errors, etc.).

Abstract:

stochasticism, "temporal and spatial correlations", but of what? One of the poorly defined parameters of the beta function? This is left for futrue work, says the conclusions, so it does not belong in the abstract as a claim about this manuscript.

p3, line 21: "inversion" layer means negative buoyancy of parcel? dT/dz >0 is what an inversion means to this reader.

p4, cloud base and air parcel source are both mentioned with no indication of how they are defined or chosen. ine 10: w* PBL should be PBL w*

p5: Beta PDF is not really a probability distribution. Clarify that it is a distribution in the height domain. Lines 7-8: how are both entrainment and detrainment profiles derived

from an assumed mass flux profile? Seems like an underdetermined problem.

p5, line 17-18: "set the vertical distribution of heat and mass" Huh? It is just a mass flux, right? Then tendencies of all scalars flow from there in the usual way.

p5, line 25-26, Fig. 2: the beta distribution has TWO parameters on [0,1]. What is being shown here exactly?

p6, line 1: Who sets the "average height of mass flux maximum", the user or the scheme via its parameters? How exactly is a beta for fitted? This is all quite unclear, more touting of supposed virtues than explaining of algorithms.

p7, are equations (4-5) redundant with (12-13) below? Units appear to be (workfunction/time) in (4), but (pressure/time) in (5). Is there something missing?

Eqs. (4) and (12) make clear that tau-bl is the STRENGTH of this term, while its temporal structure is the derivative of the BL Tv (6 hours, for solar heating effects in a 12-hour daytime). So although it naively appears one is specifying a timescale with the symbol tau-bl, it is really the maginitude of a temporal quadrature component that inherits its delay timescale from the frequency-weighted frequency spectrum of partial tendencies of Tv in a boundary layer (whose upper bound pb's definition is incidentally not given). I can probably imagine some of the contortions of logic behind this choice, but let's not pretend one is specifying a delay time, like a convection organization timescale which is more what one observes as the reason for the delay of rainfall into afternoons.

page 12, eq (11): this must be SATURATION hbar, not hbar, correct? (so that line 22 is incorrect in words)

p14, lines 20-22. Clarify how three (out of 4) Beta parameters are sufficient to define profiles of detrainment AND entrainment. Do you mean that, for a fixed entrainment profile (constant), a detreinment profile is uniquely defined from the outcome (Beta-shaped MF profile)? That I could see, if the source of the 4th parameter of the beta function were stated. Is it part of the closure, somehow?? This is frustratingly ambiguous about the information flow.

Despite the frustration of ambiguities, the brevity of the text is appreciated. These are clearly authors with important real coding work to do, above and beyond and arguably more important than writing papers. Still, this really should be improved to at least a point of clarity.

The English could use a polishing edit as well. I did not enter all the typos and word misuses that I found, I too have other jobs.

---

## Referee Comment (RC2) · Anonymous Referee #2 · 17 May 2020

Review of manuscript "The GF Convection Parameterization: recent developments, extensions, and applications" by Saulo Freitas et al.

In this paper the authors extended various aspects of the Grell-Freitas convection scheme. These include using a trimodal representation of shallow, congestus and deep convection, inclusion of a non-equilibrium closure to account for boundary layer forcing to better represent the diurnal cycle of convection, and the use of three pdfs for normalized mass flux profiles. In addition, the microphysics in convective updrafts is extended to include ice phase and associated latent heat release. Both single column and GCM simulations are performed to evaluate these changes. The results are quite

interesting. However, the presentation has much to be improved. A serious effort is needed to fix many sloppy descriptions of the GF updates. A major revision is required before I can recommend it for publication.

Major comments: 1. In several places the text was almost identical to text from another paper of the authors. While I believe this is unintentional and will refrain from calling it "self-plagiarism", it does reflect the sloppiness of the writing. For example, In the abstract of Freitas et al. (2018): "Recently, we extended the scheme to a trimodal spectral size distribution of allowed convective plumes to simulate the transition among shallow, congestus, and deep convection regimes. In addition, the inclusion of a new closure for nonequilibrium convection resulted in a substantial gain of realism in the model representation of the diurnal cycle of convection over the land." In the abstract of current manuscript: "The parameterization has been extended to a trimodal spectral size to simulate the interaction and transition from shallow, congestus and deep convection regimes. Another main new feature is the inclusion of a closure for nonequilibrium convection that resulted in a substantial gain of realism in the simulation of the diurnal cycle of convection, mainly associated with boundary layer forcing over the land." Lines 7-9 on page 3 of this manuscript: "Each of the modes is distinguished by different lateral entrainment rates that strongly control its vertical depth and, consequently, the height of the main detrainment layers." Lines 10-11 from bottom on page 1268 of Freitas et al. (2018): "Each of the modes is distinguished by different lateral entrainment rates that strongly control its vertical depth and, consequently, the height of the main detrainment layers." Such a practice of copy-and-paste from one paper to another is unacceptable. 2. Many specifics are missing in the description of the GF updates. Providing accurate information is important since a main objective of such work is to document the changes of physical schemes for interested readers. For example, P. 5, L5-8. "The mass flux profiles are given by a Beta PDF, statistically representing the normalized statistical average mass flux of deep and congestus convection in a grid box. The effective vertical entrainment rate and detrainment rate profiles are derived from these mass flux profiles." Please provide the beta pdf in the form of a

mathematical equation. Also, provide the equations for entrainment and detrainment. There are a number of such omissions. 3. In the abstract, authors states that one of recent extensions is in cloud microphysics: "Finally, the cloud microphysics has been extended to include the ice phase to simulate the conversion from liquid water to ice in updrafts with resulting additional heating release, and the melting from snow to rain within a user-specified melting vertical layer." However, there is no analysis, no figure, and no conclusion about the impact and performance of this change. If the impact is significant, please show it. 4. The authors showed the performance of GF shallow scheme only with a 2-day model simulation. Are there any observations to evaluate the shallow cumulus simulation? The mass flux shows that shallow cumulus can reach to 5.5km height, is it reasonable? Authors list three shallow convection closures in the manuscript. However, it is not clear that what closure is actually used in GF shallow scheme. What are the performance differences among these closures? A figure showing the performance of each option would be desirable. 5. In the single column model evaluation (Figs. 7&8), the authors should evaluate the simulated heating and drying tendency with available observations, for example, the analyzed diabatic heating (Q1) and drying rate (Q2) over the sounding domain during TWP-ICE described by Xie et al (2010). 6. Trimodal formulation is based on the observational analysis for tropical environment (Johnson et al., 1999). Is there any observational analysis for middle latitudes that supports this classification of convective modes? Is the scheme sensitive to the choice of entrainment rate for three mode of convection? It would be helpful to discuss this in the context of the full spectral representation recently used by Song and Zhang (2018) and Baba (2019).

Minor comments: 1. The calculation of cloud work function (CWF) using Equation (12) is problematic. The equation (11) shows that CWF is in units of m-2s-2, which is equivalent to Jkg-1. However, equation (12) shows that CWF is in units of kg2m-4s-2. The problem is that CWF~gdz in equation (11), however, CWF~-rho*dp in equation (12). Based on the hydrostatic equation (dp=-rho*gdz), the equation (12) should be divided by air density instead of being multiplied by air density. 2. Fig. 1: Is this figure diagnosed using GF from a high-resolution simulation? which shallow scheme is used? How did the authors calculate mass flux in units of kg m-2s-2 in Fig.1? Authors show mass flux in several figures but with three different units: kg m-2s-2(Fig.1), m s-1(Fig.3), kg m-2s-1 (Figs. 5 and 6). Fig.9 even does not provide the units of mass flux. Authors should clarify this and use the same correct units for mass flux so that the readers can easily compare and understand these figures. 3. P.5, L10-11. "For congestus, the closures BLQE and based on W* described in Section 2.1.1 are available, besides the instability removal using a prescribed time scale." You mean eqs. (1) and (2)? If so, state explicitly. Also, in these equations no instability removal timescale is involved. Please clarify and be specific. 4. Figure 3. What is the shading on the left and dashed lines on the right? I assume they are standard deviations. But the authors should not leave the guesswork to the readers. 5. Page 6, lines 19-21: reference should be provided. Is there any difference in diurnal cycle of convection between land and ocean regimes? 6. P. 7, L8. How does the partition vary in the mixed phase temperature range (250.16, 273.16)? 7. Page 8, line 31: "GF slightly underestimates the heavy precipitation in the active monsoon period". The underestimation of about 30% is not a slight underestimation for me. 8. P. 9, L30: Authors explain the convective cooling near cloud top by the evaporation of detrained liquid water at cloud tops. Since the cloud liquid water is detrained into environment, its evaporation cooling in the environment should not be considered convective cooling. It is usually treated as grid-scale evaporation cooling in the model. So why is there convective cooling near the cloud top? OR GF counts it differently? 9. Figure 2 shows the updraft mass flux with cloud base at model level 5 and cloud top at level 50. Why large mass flux exists below cloud base (model level 1-5)? 10. What does the dash line mean in Figure 3? 11. Figure 6. The caption says that downdraft mass flux is in green, however, there is no green line in Figure 6. It shows shallow cumulus can reach to 600hPa. Again, are there cloud observations (cloud depth, fraction) to qualitatively evaluate the simulations? 12. Fig. 5. Add a line for shallow mass flux. 13. Figure 13. It's difficult to discern much useful information from this figure. It would be better to plot the 24-h phase dial. (e.g. Fig.9

and Fig.12 in Xie et al. 2019). Also, the results of one month (January 2016) are not enough. It can be easily done with multi-year data. How is the performance of model simulation for summer months? 14. Please insert space between an equation and its number. 15. The font size in 3.2 (page10) is different to other parts in the rest of paper. 16. How is the cloud base determined for shallow, congestus and deep convection? 17. Page 12, line 7: what does "Each forecast day comprised a 120-h time integration" mean?

Refs: Baba, Y. (2019). Spectral cumulus parameterization based on cloud‐re-solving model. Climate Dynamics, 52, 309– 334. Song, X., and G. J. Zhang, 2018: The roles of convection parameterization in the formation of double ITCZ syndrome in the NCAR CESM: I. Atmospheric processes. Journal of Advances in Modeling Earth Systems, 10, https://doi.org/10.1002/2017MS001191 Xie, S., T. Hume, C. Jakob, S. A. Klein, R. B. McCoy, and M. Zhang, 2010: Observed large-scale structures and diabatic heating and drying profiles during TWP-ICE. J. Climate, 23, 57–59, https://doi.org/10.1175/2009JCLI3071.1. Xie et al., 2019: Improved diurnal cycle of precipitation in E3SM with a revised convective triggering function. Journal of Advances in Modeling Earth Systems, 11, 2290–2310.

---

## Author Comment (AC1) · 13 Feb 2021

**Reviewer 1:**
Journal: GMD
> Title: "The GF Convection Parameterization: recent developments, extensions, and applications" by Saulo R. Freitas et al.
> MS No.: gmd-2020-38

Anonymous Referee #1
We thank the reviewer for his/her insightful and helpful comments. The paper is now much improved by his/her comments and corrections. The reviewer's comments are in blue color.

("general comments")
This manuscript surveys some aspects of the latest version of the GF convection scheme, whose code is offered in an open source repository. I suppose the world1 is fortunate to have a narrative record from busy hard-working developers about what they have been doing. But I found it fairly unclear, despite my appetite for understand- ing it, and in places a bit congratulatory or justifying where the same column inches could be spent on more information. Major revision is needed if this is meant to reach the standards of a formal "scientific publication" in a traditional sense. Since a lot of prior knowledge of the subject is assumed, and since the text has little pretense to a physical rationale for the algorithms being described, the paper could be very short and to the point: what is the informatic mapping from inputs and assump- tions, to internal variables, to outputs? Unfortunately, a careful read of this manuscript is required to fish out many of these mappings and assumptions, and then many details are left ambiguous, for instance about what kinds of choices a user must make (and the authors DID make in the examples presented).
In short, it could use a tightening-up in style, and a completeness check for crucial details. Here is an indented bulleted list of the inputs, internals, and outputs as I see them. The formatting system seems to mess them up, I find, but this is what I offer as an unpaid volunteer; the reader can guess the hierarchy.
Inputs:
- T and q and wind and tracer profiles
- the Tv tendency averaged over the PBL from a separate BL scheme
- an aerosol-related input to autoconversion efficiency in the height domain Assumed parameters:
- what level to start entraining parcels/plumes from, never explained
- 3 "initial" entrainment rates {2, 0.9, 0.1} /km , buried in different places
- something about the undetermined 4th parameter of the Beta profile curve
- closure choices for shallow convection - for 3 different approaches; and rules for how to select or combine them
- closure choices for middle and deep convection -assumed timescale for convection to damp the work function - the strength of a diurnal delay trick, misleadingly cast as a timescale tau_b
- temperatures for freezing and melting (melting should not begin at -3C, should it?)
Intermediate variables:
- 4 parameters of the parametric steady-state mass flux profiles for each of the 3 kinds of updraft plumes - 2 params: how to map Beta's [0,1] domain into levels, z, log-p, or p (various figures use all four)

- 2 params: shape of the profile - cast as the altitude of the maxiumum? - cast as the "mean cloud base"? - inverted from the closure somehow, along with a detrainment profile? - is an entrainment profile different from constant also backed out?
- All of the above for downdrafts too? downdrafts are mentioned, but never specified
- Finite area fraction profiles for minor "scale aware" adjustments of eddy flux from MF?
- Special assumptions for momentum vs. other tracers? (reflecting "pressure effects")
Outputs:
- profiles of tendencies of state variables (or the final result of successively updated profiles?)
- surface rainfall rate
Unfortunately, the manuscript is not clear at this level. It should be, since that seems to be its one job.

The manuscript was completely rewritten following the concerns and recommendations of the two reviewers. Thanks for taking your time to advise us on how to improve it. We understand this version is now much clear, complete, and polished; we hope it is now acceptable for your and GMD standards.

("specific comments")
Allowing for its very high-level narrative intent, the paper still needs major clarifications.
At the outset, the gross characterizations of the GF scheme's type and intent should be given. It is a steady-state updraft plume model, evidently with no downdrafts - oh wait, page 5 line 19 mentions them, presumably as inverted saturated plumes of the same kind? Precipitation removal is not mentioned, except to say that it can be made "aerosol aware" via an autoconversion of some sort, and can optionally be disabled entirely for the plume with 2/km initial entrainment rate. Momentum flux is mentioned in the abstract, but never in the paper, so presumably all quantities are simply transported identically, based on plume mass flux, is that correct? Or is there some pressure treatment for momentum? How does the scheme define its cloud base, for one or all plumes (LCL of surface air?) and its parcel properties (mixture up to that LCL, or does it entrain its way through the subcloud layer?) How about cloud top - is it the highest or lowest level of neutral buoyancy for the initial entraining-only instability- probing parcel calculation with its specified entrainment rate? (2, 0.9, and 0.1 /km; these three numbers are buried in the text).
We detail the information about how the parameters for the three types of convection are defined and improved the description Please, take a look at the new section 2.

The 3 plumes are said to be successive in time; does this mean the scheme updates the state profile within a call? Merely for its own internal accounting, such that it re- turns only one tendency profile to the mother model, or is this entangled inseparably with a model's time-marching scheme? Must the host model use the same conserved variables? (presumably with some fixed constant reference values for Cp, Lv, etc.?)
The three convective types are called successively in the same time step. At his point they do not update the state successively, since we were not able yet to make a comprehensive assessment of the model performance using this approach. So far, they all use the same state of the host model and the net feedback is simply the sum of three individual tendencies produced by each one. The scheme provides tendencies for temperature, water vapor and condensates mixing ratio.

Once the cloud base and top are determined, these are somehow (it is very ambiguous) used to fit a Beta Distribution in the height (or level? or log-pressure? or pressure? Figures include all four) domain, as the mass flux (MF) profile. Presumably that defines the eddy flux of all scalars and the condensation rate? Although lines 1-7 of Page 6 make it sound like the heating and moistening rates can ALSO be specified, or that they TOO can be made smooth because MF is, that is not how tendencies work out for a smoothly varying mass flux through a sharp inversion. Beta is a two-parameter curve family on [0,1] so we are left to guess how THREE of Beta's FOUR parameters are set by (1) cloudbase cb, perhaps the LCL of some undefined parcel origination level?, and (2) one of its neutral buoyancy levels?, and (3a) perhaps something that is different from cb called "average cloud base" (line 21 of page 5) or (3b) "the average height of mass flux maximum" (line 1 of page 6)? Very unclear. Read the code, I suppose. Beta's FOUR parameters are: the mapping of [0,1] onto height, and Beta's two param- eters, leaving only one parameter "free", to be defined by ?the closure? From Fig. 3 (why are there 3 curves?), it appears that the [0,1] domain of Beta is not the same as some clipping or masking at cb and ct. From the mass flux profile, entrainment and detrainment are backed-out, although how TWO profiles are backed out of ONE is un- clear. Is the buoyancy of the new entraining-detraining plume used to revisit the initial cloud base and top, or are these retained once fixed by the constant-entrainment-rate parcel buoyancy?

Quite unclear, all of this.

The section 2 was completely redone to make clear the formulation for defining the cloud properties, entrain/detrainment rates and the normalized mass flux profile (Beta function). We also added a section for a better understanding of how the PDFs function, and how this might be used for stochastic applications or training methods.

Freezing and melting are now accounted for, which seems fine and necessary, but for some reason melting begins at -3C, which makes no physical sense to me.

Done.

("technical corrections": typing errors, etc.).

Abstract:

stochasticism, "temporal and spatial correlations", but of what? One of the poorly de- fined parameters of the beta function? This is left for futrue work, says the conclusions, so it does not belong in the abstract as a claim about this manuscript.

This feature is no longer present in the Abstract.

p3, line 21: "inversion" layer means negative buoyancy of parcel? dT/dz >0 is what an inversion means to this reader.

This expression is now clarified, thanks.

p4, cloud base and air parcel source are both mentioned with no indication of how they are defined or chosen. ine 10: w* PBL should be PBL w*

Done, thanks.

p5: Beta PDF is not really a probability distribution. Clarify that it is a distribution in the height domain.

We are following the nomenclature given in https://en.wikipedia.org/wiki/Beta_distribution, and using the PDF defined in this web page.

Lines 7-8: how are both entrainment and detrainment profiles derived from an assumed mass flux profile? Seems like an underdetermined problem.
The determination of the entr/detrainment profiles is now better informed. Please, take a look at the new Section 2.

p5, line 17-18: "set the vertical distribution of heat and mass" Huh? It is just a mass flux, right? Then tendencies of all scalars flow from there in the usual way.
That expression is no longer present in the manuscript. However, changes in the Beta parameter and the height of maximum normalized mass profiles imply changes in the tendencies. See a short discussion at the end of the new Section 2.3.

p5, line 25-26, Fig. 2: the beta distribution has TWO parameters on [0,1]. What is being shown here exactly?
We show the normalized mass flux profile in terms of the beta parameter for a given cloud base, cloud top, and maximum mass flux height. The cloud base may be determined by the boundary layer height (shallow and congestus convection) or through determination where the level of free convection is located. The assumed statistically averaged cloud top for deep, congestus, or shallow convection is determined by environmental conditions in addition to the assumed average characteristic size, given by an initial gross entrainment rate. The height of maximum mass flux is also given by environmental conditions (explained in section 2). An option exists to also supply this level as an input variable. Please, revisit Section 2 for a new, more detailed explanation.

p6, line 1: Who sets the "average height of mass flux maximum", the user or the scheme via its parameters? How exactly is a beta for fitted? This is all quite unclear, more touting of supposed virtues than explaining of algorithms.
Please, revisit Section 2 for a new, more detailed explanation.

p7, are equations (4-5) redundant with (12-13) below? Units appear to be (workfunc- tion/time) in (4), but (pressure/time) in (5). Is there something missing?
Equations 4-5 are the original formulation by Bechtold et al (2014), whereas Equations 12-13 represent the adaptation of those equations in the GF scheme. Section 2.3 was rewritten to make this clear. Thanks.

Eqs. (4) and (12) make clear that tau-bl is the STRENGTH of this term, while its tempo- ral structure is the derivative of the BL Tv (6 hours, for solar heating effects in a 12-hour daytime). So although it naively appears one is specifying a timescale with the symbol tau-bl, it is really the maginitude of a temporal quadrature component that inherits its delay timescale from the frequency-weighted frequency spectrum of partial tendencies of Tv in a boundary layer (whose upper bound pb's definition is incidentally not given). I can probably imagine some of the contortions of logic behind this choice, but let's not pretend one is specifying a delay time, like a convection organization timescale which is more what one observes as the reason for the delay of rainfall into afternoons.

The reference below describes the original development of this approach:
Bechtold, P., N.et al.: Representing Equilibrium and Nonequilibrium Convection in Large-Scale
Models. J. Atmos. Sci., 71, 734–753, doi: 10.1175/JAS-D-13-0163.1, 2014.

page 12, eq (11): this must be SATURATION hbar, not hbar, correct? (so that line 22 is incorrect
in words)
Done, thanks.

p14, lines 20-22. Clarify how three (out of 4) Beta parameters are sufficient to define profiles of
detrainment AND entrainment. Do you mean that, for a fixed entrainment profile (constant), a
detreinment profile is uniquely defined from the outcome (Beta- shaped MF profile)? That I
could see, if the source of the 4th parameter of the beta function were stated. Is it part of the
closure, somehow?? This is frustratingly ambiguous about the information flow.
Please, revisit Section 2.2 for a clear discussion about how those profiles are derived.

Despite the frustration of ambiguities, the brevity of the text is appreciated. These are clearly
authors with important real coding work to do, above and beyond and arguably more important
than writing papers. Still, this really should be improved to at least a point of clarity. The English
could use a polishing edit as well. I did not enter all the typos and word misuses that I found, I
too have other jobs.

Additional English proofreading was performed. Also, during production, Copernicus
Publications applies typesetting and language copy-editing. We understand that the final version
of the manuscript will have an acceptable level of language quality and correction.

---

## Author Comment (AC2) · 13 Feb 2021

**Reviewer 2:**

Journal: GMD
> Title: "The GF Convection Parameterization: recent developments, extensions, and applications" by Saulo R. Freitas et al.
> MS No.: gmd-2020-38

Anonymous Referee #2
We thank the reviewer for his/her insightful and helpful comments. The paper is now much improved by his/her comments and corrections. The reviewer's comments are in blue color.

In this paper the authors extended various aspects of the Grell-Freitas convection scheme. These include using a trimodal representation of shallow, congestus and deep convection, inclusion of a non-equilibrium closure to account for boundary layer forcing to better represent the diurnal cycle of convection, and the use of three pdfs for normalized mass flux profiles. In addition, the microphysics in convective updrafts is extended to include ice phase and associated latent heat release. Both single column and GCM simulations are performed to evaluate these changes. The results are quite interesting. However, the presentation has much to be improved. A serious effort is needed to fix many sloppy descriptions of the GF updates. A major revision is required before I can recommend it for publication.

Thanks again for your work on reviewing the manuscript and making recommendations. We did our best to accomplish all of them and hope it is now suitable for publication.

**Major comments:**

1. In several places the text was almost identical to text from an- other paper of the authors. While I believe this is unintentional and will refrain from calling it "self-plagiarism", it does reflect the sloppiness of the writing. For example, In the abstract of Freitas et al. (2018): "Recently, we extended the scheme to a tri- modal spectral size distribution of allowed convective plumes to simulate the transition among shallow, congestus, and deep convection regimes. In addition, the inclusion of a new closure for nonequilibrium convection resulted in a substantial gain of real- ism in the model representation of the diurnal cycle of convection over the land." In the abstract of current manuscript: "The parameterization has been extended to a tri- modal spectral size to simulate the interaction and transition from shallow, congestus and deep convection regimes. Another main new feature is the inclusion of a closure for nonequilibrium convection that resulted in a substantial gain of realism in the simulation of the diurnal cycle of convection, mainly associated with boundary layer forcing over the land." Lines 7-9 on page 3 of this manuscript: "Each of the modes is distinguished by different lateral entrainment rates that strongly control its vertical depth and, consequently, the height of the main detrainment layers." Lines 10-11 from bottom on page 1268 of Freitas et al. (2018): "Each of the modes is distinguished by different lateral entrainment rates that strongly control its vertical depth and, consequently, the height of the main detrainment layers." Such a practice of copy-and-paste from one paper to another is unacceptable.

Those expressions are no longer present and sorry about that.

2. Many specifics are missing in the description of the GF updates. Providing accurate information is important since a main objective of such work is to document the changes of physical schemes for interested readers. For example, P. 5, L5-8. "The mass flux profiles are given by a Beta PDF, statistically representing the normalized statistical average mass flux of deep and congestus convection in a grid

box. The effective vertical entrainment rate and detrainment rate profiles are derived from these mass flux profiles." Please provide the beta pdf in the form of a mathematical equation. Also, provide the equations for entrainment and detrainment. There are a number of such omissions.

We rewrote the description of the use of the PDF's and gave much more details, including a description of how this approach could be used for stochastics and/or tuning for operational applications. Section 2.2 is a full description of the PDF approach of the GF scheme as it is available on github, and used operationally in the RAP forecast system at NCEP/U.S.

3. In the abstract, authors states that one of recent extensions is in cloud microphysics: "Finally, the cloud microphysics has been extended to include the ice phase to simulate the conversion from liquid water to ice in updrafts with resulting additional heating release, and the melting from snow to rain within a user-specified melting vertical layer." However, there is no analysis, no figure, and no conclusion about the impact and performance of this change. If the impact is significant, please show it.

The main feature in extending to include the ice phase is additional heating at upper levels associated with the phase change from liquid to ice. We did not see a significant impact overall. However, including this extension makes the formulation physically more realistic.

4. The authors showed the performance of GF shallow scheme only with a 2-day model simulation. Are there any observations to evaluate the shallow cumulus simulation? The mass flux shows that shallow cumulus can reach to 5.5km height, is it reasonable? Authors list three shallow convection closures in the manuscript. However, it is not clear that what closure is actually used in GF shallow scheme. What are the performance differences among these closures? A figure showing the performance of each option would be desirable.

R. The figure 1 was replaced by the results of the SCM run over the Amazon basin. The text was completely rewritten (section 3.2), and the Figure 8 shows the results.

5. In the single column model evaluation (Figs. 7&8), the authors should evaluate the simulated heating and drying tendency with available observations, for example, the analyzed diabatic heating (Q1) and drying rate (Q2) over the sounding domain during TWP-ICE described by Xie et al (2010).

We added Q1 and Q2 profile discussions. Shape is usually very similar, but magnitudes are somewhat different, even for a comparison of profiles calculated over the sounding domain from Xie et al (2010) and observed profiles supplied by the SCM.

6. Trimodal formulation is based on the observational analysis for tropical environment (Johnson et al., 1999). Is there any observational analysis for middle latitudes that supports this classification of convective modes? Is the scheme sensitive to the choice of entrainment rate for three mode of convection? It would be helpful to discuss this in the context of the full spectral representation recently used by Song and Zhang (2018) and Baba (2019).

We don't think that congestus convection is limited to the tropical regions. We modified the sentence in the manuscript to clarify this a little. In addition, while we are not representing a smooth spectral representation of all convective cloud types as is the intention in Song and Zhang (2018) and Baba (2019), the PDFs used here are to represent a statistical average of three cloud types, but that does not mean they are always the same size. A PDF for deep convection may represent several cloud types. The top of those cloud type is given and the location of the maximum mass flux in the vertical, but that does not mean that cloud types contribute that are not reaching the top of the PDF. A similar assumption is

valid for congestus and shallow convection. This is also why mass flux of shallow convection may reach up to 550mb.

Minor comments:

The calculation of cloud work function (CWF) using Equation (12) is problematic. The equation (11) shows that CWF is in units of m-2s-2, which is equivalent to Jkg-1. However, equation (12) shows that CWF is in units of kg2m-4s-2. The problem is that CWF~gdz in equation (11), however, CWF~-rho*dp in equation (12). Based on the hydrostatic equation (dp=-rho*gdz), the equation (12) should be divided by air density instead of being multiplied by air density.

Sorry for the typo and thanks for point it out.

Fig. 1: Is this figure diagnosed using GF from a high-resolution simulation? which shallow scheme is used? How did the authors calculate mass flux in units of kg m-2s-2 in Fig.1? Authors show mass flux in several figures but with three different units: kg m-2s-2(Fig.1), m s-1(Fig.3), kg m-2s-1 (Figs. 5 and 6).

The figure 1 was removed. The units in Figure 3 are wrong, they are kg $m^{-2}$ $s^{-1}$. The units in Figs 5 and 6 are correct.

Fig.9 even does not provide the units of mass flux. Authors should clarify this and use the same correct units for mass flux so that the readers can easily compare and understand these figures.

The figure was replaced and the total mass flux from the 3 modes is now shown in units of kg $m^{-2}$ $s^{-1}$.

3. P.5, L10-11. "For congestus, the closures BLQE and based on W* described in Section 2.1.1 are available, besides the instability removal using a prescribed time scale." You mean eqs. (1) and (2)? If so, state explicitly. Also, in these equations no instability removal timescale is involved. Please clarify and be specific.

The text reads now: "For congestus, the closures BLQE (Eq. 1) and based on W* (Eq. 2) described in Section 2.1.1 are available, besides the instability (measured as the cloud work function) removal using a prescribed time scale of 1800 seconds (see Section 2.3 for further details)."

Figure 3. What is the shading on the left and dashed lines on the right? I assume they are standard deviations. But the authors should not leave the guesswork to the readers.

As asked by the reviewers, the Section 2 was completely redone, and this figure no longer is present in manuscript.

Page 6, lines 19-21: reference should be provided. Is there any difference in diurnal cycle of convection between land and ocean regimes?

A reference was added. A brief discussion about the diurnal cycle of convection in both regimes is present in Section 3.3.

P. 7, L8. How does the partition vary in the mixed phase temperature range (250.16, 273.16)?

R. The partition is now explicitly informed in 2.4. Thanks for asking.

Page 8, line 31: "GF slightly underestimates the heavy precipitation in the active monsoon period". The underestimation of about 30% is not a slight underestimation for me.

Rewrite in paper as:
Compared to single location precipitation data the maximum amount appears underestimated.
However, the average when using GF is over an area that covers a much larger domain.

P. 9, L30: Authors explain the convective cooling near cloud top by the evaporation of detrained liquid water at cloud tops. Since the cloud liquid water is detrained into environment, its evaporation cooling in the environment should not be considered convective cooling. It is usually treated as grid-scale evaporation cooling in the model. So why is there convective cooling near the cloud top? OR GF counts it differently?

The reviewer is correct, that the resolved microphysics will evaporate the detrained cloud water, which may lead to cooling. However, in GF impacts from convection are from both, detrainment of water vapor, detrainment of moist static energy, and subsidence impacting both. Since much water vapor is detrained this will lead to cooling, especially for shallow and congestus clouds, since the amounts of water vapor are larger, and the subsidence impact on moist static energy is smaller. The equation below shows those impacts:

$$\frac{\partial T(k)}{\partial t} = \frac{1.}{cp} \varrho[h(z)] * m_{b(CU)} - \frac{L_v}{cp} \varrho[q(z)] * m_{b(CU)}$$

Here the $\varrho$ is the change of moist static energy ($h$) or water vapor ($q$) per unit mass, and $m_{b(CU)}$ is the cloud base mass flux for deep, congestus, or shallow convection.

Figure 2 shows the updraft mass flux with cloud base at model level 5 and cloud top at level 50. Why large mass flux exists below cloud base (model level 1-5)?

The figure 2 was redone. The cloud base height is about ~ 1.2 km and the mass flux increases from that level.

What does the dash line mean in Figure 3?

This figure is no longer in this manuscript version.

Figure 6. The caption says that downdraft mass flux is in green, however, there is no green line in Figure 6. It shows shallow cumulus can reach to 600hPa. Again, are there cloud observations (cloud depth, fraction) to qualitatively evaluate the simulations?

Considering we already show the trimodal cloud characteristics in Fig 5b, and also got the AMS permission to use the observational Figure 13 from Kumar et al. 2016 as Fig 5a. We deleted the original Figure 6, and added the validation of diabatic heating source (Q1) and drying sink (Q2) in the revised manuscript.

Fig. 5. Add a line for shallow mass flux.

Following the reviewer's suggestion, a line in blue for shallow mass flux was added in Figure 5b.

Figure 13. It's difficult to discern much useful information from this figure. It would be better to plot the 24-h phase dial. (e.g. Fig.9 and Fig.12 in Xie et al. 2019). Also, the results of one month (January 2016) are not enough. It can be easily done with multi-year data. How is the performance of model simulation for summer months?

R. We included more results and discussion based on the approach shown in Xie et 2019, and the analysis was also performed for July 2015. Using this approach, we focused on the CONUS, Amazon, tropical north of Africa, and the Tropical Pacific Ocean. Please, take a look at the new Section 3.3.

Please insert space between an equation and its number.

Done, thanks.

The font size in 3.2 (page10) is different to other parts in the rest of paper.

Done, thanks.

How is the cloud base determined for shallow, congestus and deep convection?

This information is now much more detailed in the Section 2.

Page 12, line 7: what does "Each forecast day comprised a 120-h time integration" mean?

The text now reads: "Each forecast day covered a 120-h time integration, with output available every hour."

Refs:

Baba, Y. (2019). Spectral cumulus parameterization based on cloud resolving model. Climate Dynamics, 52, 309– 334.

Song, X., and G. J. Zhang, 2018: The roles of convection parameterization in the formation of double ITCZ syndrome in the NCAR CESM: I. Atmospheric processes. Journal of Advances in Modeling Earth Systems, 10, https://doi.org/10.1002/2017MS001191

Xie, S., T. Hume, C. Jakob, S. A. Klein, R. B. McCoy, and M. Zhang, 2010: Observed large-scale struc- tures and diabatic heating and drying profiles during TWP-ICE. J. Climate, 23, 57–59, https://doi.org/10.1175/2009JCLI3071.1. Xie et al., 2019:

Improved diurnal cycle of precipitation in E3SM with a revised convective triggering function. Journal of Ad- vances in Modeling Earth Systems, 11, 2290–2310.

---

## Author Comment (AC3) · 13 Feb 2021

Dear Dr. Kerkweg. We followed your requirements, and the new manuscript should accomplish all of them. Saulo Freitas

---

## Author Response (AR2)

Journal: GMD

Title: "The GF Convection Parameterization: recent developments, extensions, and applications" by Saulo R. Freitas et al.

MS No.: gmd-2020-38

Anonymous Referee #1

The manuscript is much improved in response to first reviews. It is important that schemes have their documentation papers, so I recommend publication (whatever that means for copernicus journals like GMD, I remain confused by this whole scholarship model).

A few points of clarification will help with readability, but I focus my comments and suggestions only on the Abstract, since any reader who will take the time to wade into the internal structure will have committed the time to parse the subtleties.

We thank the reviewer for his/her insightful and helpful comments. The paper is now much improved by his/her comments and corrections. The reviewer's comments are in blue color.

Line 11 "Abstract: We detail recent developments" — it will be clearer if the phrase AND OPTIONS is added. There are multiple treatments, for instance multiple closures and now (line 16) "we also added a new closure". In the first reading I was confused about how these closures all connect or relate. Now I understand that these are various options within the code, all of which are being described here for completeness. In the applications section 3 (which I did not read carefully), the parameter values and choices should be specified, preferably in a Table. There are no tables currently.

Done.

Line 14: "we assume that Probability Density Functions (PDFs) can be used to characterize the vertical mass flux profiles..." This is incorrect and confusing language! Probability is not used. Instead, the Beta Function is assumed to be the shape of mass flux profiles. While the beta function is a normalized function, such that it can be sometimes used in probability theory, the functional form here is NOT a probability: rather, mass flux $M(p)$ for each plume type is assumed to be a beta function of the pressure coordinate. The symbol Z is used for mass flux rather than M, for unclear reasons. The symbol r is used for a (relative) pressure (5), for unclear reasons. So this rather strong assumption that $M(p)$ is smooth is somewhat obfuscated as a $Z(r)$ smoothness assumption. In any case, it seems quite a drastic physical assumption, as stable layers and other features of stratification often induce blips in mass flux profiles through entrainment and detrainment (for instance, buoyancy sorting). That process is important in making the stratification moist adiabatic in convecting regions, ironing out the kinks and inversions in a sounding. None of the figures shown indicate whether this key job of buoyant convection is actually performed under this profile assumption. With 3 smooth profiles and time intermittency it probably works itself out, but that physical assumption remains a debatable one, and mustn't be covered up by calling it a probability density assumption!

We thank the reviewer for the comments. We changed the expression, and now we are naming it as 'Beta Function.' The text was altered throughout to reflect the new term.
* * *
Journal: GMD

Title: "The GF Convection Parameterization: recent developments, extensions, and applications" by Saulo R. Freitas et al.

MS No.: gmd-2020-38

Anonymous Referee #2

2nd review of "The GF Convection Parameterization: recent developments, extensions, and applications" by S. Freitas et al.
The manuscript is significantly improved although there are still areas that need further improvement. I have a few additional comments.
We thank the reviewer for his/her insightful and helpful comments. The reviewer's comments are in blue color.

Major comments:
1. The authors added 4 more figures (Figs. 14-17), all on diurnal cycle. Unfortunately, this makes the application section extremely lopsided, with 10 out of 17 figures on various aspects of the precipitation diurnal cycle associated with the use of boundary layer cloud work function generation. As such, the current title is not quite fitting. Alternatively, the authors can consolidate the diurnal cycle part and make the material more balanced.
We understand that including the diurnal cycle closure is a significant advance in this parameterization. That explains the larger number of figures and discussions. However, the model version still has several other new features that justify the current title.

2. The writing is much improved. However, there are still many places where the text reads quite rough. I suggest the authors go over the text thoroughly and correct the errors/misuse of words and polish the writing.
Additional English proofreading was performed. Also, during production, Copernicus Publications applies typesetting and language copy-editing. We understand that the final version of the manuscript will have an acceptable level of language quality and correction.

Minor comments:
1. P. 5, L12-13. By "thermal inversion" do you mean you still look for dT/dz>0 near 500 hPa to define congestus cloud top as you do for shallow convection? If so, this would be unrealistic, as in the free troposphere on a spatial scale of a GCM grid box size it would be hard to find dT/dz>0.
The thermal inversion is defined following the two criteria below:
- the first derivative (del T/del Z) must have a local maximum
- the absolute value of the second derivative must be zero.

2. Eq. (16), what is the value of tau_BL?
Sorry about that. It is now defined.

3. It appears zt (cloud top height in eq. (15)) is defined by the neutral buoyancy and/or inversion layer. But in Figs. 11 and 12, there are negative values of total CWF in the composite diurnal cycle. Is this largely from the negative buoyancy contribution below the level of free convection? If so, it's worth mentioning. If not, where do the negative contributions come from? The global average CWF is very small; there must be large negative contributions from mid- and high latitudes.
It seems to be related to the negative buoyancy contribution below the level of free convection. That is included in the text.

4. Regarding Fig. 12, it seems the composite over all grid points globally (plus land and ocean separately). This may not be meaningful since most of the grid points outside the tropics will have stable atmosphere. I suggest dropping this figure.
Thanks for the suggestion, but we think it is still beneficial to report the characteristics of the time evolution of the cloud work function and the boundary layer production.

5. Figs. 8-10 and 11-12 used time inconsistently. Please use either local time or UTC, but not mixed use of both.

We believe that even using the local time and UTC, the specification is clear for the reader. Additionally, for the figures with UTC, a mark denotes when the sunrise happens, giving the reader a sense of local time.

6. Fig. 15. Left column is not labeled.
Done, thanks.

7. Subsection 2.4. I asked about the effect of including freezing heating and the authored responded, stating the effect is minimal. It should be mentioned in the manuscript. Otherwise, readers may wonder what effects this modification has.
Done, thanks.

List of a few typos/grammatic errors (there are many more):
Additional English proofreading was performed by a native English speaker. Also, during production, Copernicus Publications applies typesetting and language copy-editing. We understand that the final version of the manuscript will have an acceptable level of language quality and correction.

1. P. 8, L14. "implies, for example, in a more evenly detrainment" should be "implies, for example, a more even detrainment".
Done, thanks.

2. P. 16, L25, "averaged areal", make a correction.
Done, thanks.

L28, "does not do much". To what?
Done, thanks.

3. P. 18, L20-21. "Both simulations were not able to...". Change to "...were unable to...".
Done, thanks.